# Deep Kernel Posterior Learning under Infinite Variance Prior Weights

**Jorge Loría**[*]
Department of Computer Science
Aalto University
Espoo, Finland
`jorge.loria@aalto.fi`

**Anindya Bhadra**
Department of Statistics
Purdue University
West Lafayette, IN, USA
`bhadra@purdue.edu`

## Abstract

Neal (1996) proved that infinitely wide shallow Bayesian neural networks (BNN) converge to Gaussian processes (GP), when the network weights have bounded prior variance. Cho & Saul (2009) provided a useful recursive formula for deep kernel processes for relating the covariance kernel of each layer to the layer immediately below. Moreover, they worked out the form of the layer-wise covariance kernel in an explicit manner for several common activation functions, including the ReLU. Subsequent works have made the connection between these two works, and provided useful results on the covariance kernel of a deep GP arising as wide limits of various deep Bayesian network architectures. However, recent works, including Aitchison et al. (2021), have highlighted that the covariance kernels obtained in this manner are *deterministic* and hence, precludes any possibility of representation learning, which amounts to learning a non-degenerate posterior of a random kernel given the data. To address this, they propose adding artificial noise to the kernel to retain stochasticity, and develop deep kernel Wishart and inverse Wishart processes. Nonetheless, this artificial noise injection could be critiqued in that it would not naturally emerge in a classic BNN architecture under an infinite-width limit. To address this, we show that a Bayesian deep neural network, where each layer width approaches infinity, and all network weights are elliptically distributed with infinite variance, converges to a process with $\alpha$-stable marginals in each layer that has a *conditionally Gaussian* representation. These conditional random covariance kernels could be recursively linked in the manner of Cho & Saul (2009), even though marginally the process exhibits stable behavior, and hence covariances are not even necessarily defined. We also provide useful generalizations of the recent results of Loría & Bhadra (2024) on shallow networks to multi-layer networks, and remedy the prohibitive computational burden of their approach. The computational and statistical benefits over competing approaches stand out in simulations and in demonstrations on benchmark data sets.

## 1 Introduction

The study of deep kernel processes as the infinite-width limit of deep Bayesian neural networks can be traced back to the work of Cho & Saul (2009), and further developed by Damianou & Lawrence (2013); Wilson et al. (2016); Aitchison et al. (2021), among others. This connection is built upon the foundational work of Neal (1996), who proved under the condition of bounded variance network weights, the infinite-width limit of a shallow (one hidden layer) Bayesian neural network (BNN) is a Gaussian process (GP), whose covariance kernel depends on the choice of the nonlinear activation function. Neal (1996) studied two bounded activations explicitly: the sign function and $\tanh$, and worked out the covariance kernels in these cases: respectively the exponential and the squared exponential covariances. Subsequently, Williams (1996) obtained explicit expressions for the kernel under an error function activation. Through a standard use of what is now known as the *kernel trick*, these expressions then allow one to work on with the covariance matrix induced by the kernel at the

---

[*]Corresponding author

observation points for the purpose of nonparametric estimation and prediction under a GP model, rather than having to work with a possibly infinite-dimensional and nonlinear feature space.

The importance of the work of Cho & Saul (2009) in the deep GP literature over the early works on shallow networks is that they extend the kernel processes to *deep* architectures, which emerge as infinite width limits of deep BNNs under bounded variance weights in various architectures, such as simple deep feedforward BNN (Lee et al., 2018; de G. Matthews et al., 2018), or deep convolutional BNNs (Garriga-Alonso et al., 2018). In fact, the tensor program of Yang (2019) shows that a limiting GP would appear under an arbitrary network architecture under bounded variance priors. The work of Cho & Saul (2009) provides an explicit recursive formula relating the kernels at each layer to the layer below, so that one can evaluate the covariance matrix of the features in layer $\ell + 1$, using only the covariance matrix of the features in layer $\ell$. However, in the case of deep Gaussian processes, the covariance kernel in each layer is deterministic (non-stochastic) and the downside is that any possibility of learning the posterior distribution of the features disappears (Aitchison, 2020), as the posterior distribution of a degenerate point mass kernel is again a degenerate point mass.

To address the deterministic limit, Aitchison et al. (2021) propose deep inverse Wishart processes (DIWP), which they argue approximates neural network GPs and have convenient properties for learning a variational posterior. Before introducing DIWP they also propose deep Wishart processes (DWP), which work with a noisy version of the sample covariance matrices for finite-width DNNs, the downside being a deterministic infinite-width limit. While these approaches succeed in making the covariance kernel random through an artificial noise injection, so that the posterior is non-degenerate, these processes do not result naturally as an infinite-width limit as in Neal (1996) or Lee et al. (2018). Consequently, the choice of the artificial noise distribution could itself be considered a somewhat subjective design parameter, even though some recommendations and desiderata for such choices do exist, such as the deterministic limiting kernel is a *centrality parameter* of the distribution assigned to the random kernel, which is ensured in both DIWP and DWP.

Although deep GPs emerging as the infinite-width limit of deep BNNs are definitely interesting in their own right, there are some limitations of a Gaussian scaling limit. One is the lack of feature learning mentioned above. Second is the mean-square continuity of GPs, which means a GP prior cannot model jump discontinuities or even just localized smoothness. The third limitation is one noted by Neal (1996), that in the case of multiple outputs, GPs cannot learn the covariance structure under i.i.d. weights, as the covariance is always zero under an isotropic Gaussian limit. To address these limitations we develop $\alpha$-stable kernel processes as the infinite-width limit of deep BNNs with infinite prior variance at each layer. The realizations of $\alpha$-stable random variables are able to model jumps (Kyprianou, 2018, Section. 1.2), and their ability to learn features via a stochastic kernel is developed in this paper.

## 1.1 Prior works on infinite-width limits of BNNs under infinite variance priors and our contributions in context

The literature on the scaling limits of infinitely wide BNNs with weights that have infinite variance priors has traditionally focused on the properties of the limiting process and less on posterior inference under such prior processes (analogous to kriging under GP priors). The first non-Gaussian limit result of its kind was given by Der & Lee (2005), who proved that in a single hidden layer neural network the limiting process is sub-Gaussian with stable margins, and derived the characteristic function of the process. Recent extensions to deep feedforward networks are by Peluchetti et al. (2020), Favaro et al. (2023) and Lee et al. (2023); and to the convolutional setting by Bracale et al. (2022). To the best of our knowledge, the first computationally viable method for posterior inference under these prior processes was explored by Loría & Bhadra (2024). However, their method has two major drawbacks. The first is that the method is limited to shallow networks. The second is the computational complexity that scales exponentially in the input dimension. Specifically, with $n$ training points of dimension $I$, the computational complexity of their method is $\mathcal{O}(n^{I+2})$. Although Loría & Bhadra (2024) provided considerable evidence in one and two dimensions that a stable process prior is beneficial when the true underlying data-generating function contains jump-type discontinuities (which arise with zero probability under a GP prior), the second drawback, i.e., the exponential complexity in $I$, is especially limiting, since it makes it infeasible to apply the method in all but one or two dimensions, as also evidenced by our numerical experiments presented later in this paper. An intuitive explanation for this exponential complexity is that Loría & Bhadra (2024)

work in the *feature space* with sign activation function in the hidden layer, for which the resulting separating hyperplanes require enumeration over each of the $I$ dimensions on whether the output of the activation function is marked as 1 or 0. Enumerating over all $n$ points results in the exponential in $I$ complexity. Avoiding this combinatorial enumeration is of critical importance for large $I$.

To address these serious limitations, the current work eschews working in the feature space altogether, and focuses on kernel methods. Setting aside the issue of computational complexity for a moment, a key observation of Loría & Bhadra (2024) is that the limiting stable process has a conditionally Gaussian representation, i.e., it is a Gaussian process, conditioned on some positive $\alpha$-stable variables. To see the main idea, we introduce the necessary notations on $\alpha$-stable random variables at this point. Define $X \sim S(\alpha, \beta)$ an $\alpha$-stable random variable with index parameter $\alpha \in (0, 2]$ and skewness parameter $\beta \in [-1, 1]$, by its characteristic function: $\phi_X(t) = \mathbb{E}[\exp(\mathrm{i}tX)] = \exp\{-|t|^\alpha[1 - \mathrm{i}\beta\omega(\alpha, t)]\}$, with $\omega(\alpha, t) = \tan(\alpha\pi/2)\mathrm{sign}(t)$ for $\alpha \neq 1$, and $\omega(1, t) = -(2/\pi)\log(|t|)\mathrm{sign}(t)$ (Samorodnitsky & Taqqu, 1994, p. 5), with no closed form to the density of $X$ in general, apart from specific $\alpha$. A remarkable property of $\alpha$-stable variables is that the $\alpha = 2$ case is Gaussian, with the usual Gaussian limit for the scaled sum established by the classical central limit theorem (CLT); but they possess infinite variance for $\alpha < 2$, which does not admit a Gaussian scaling limit due to an inapplicability of the classical CLT. The generalized CLT (Gnedenko & Kolmogorov, 1968), that still applies, gives the non-Gaussian scaling limit in this case, and is stated in Appendix A. Two cases of interest are: symmetric $\alpha$-stable random variables, which have $\beta = 0$; and positive $\alpha$-stable, which requires $\alpha < 1$ and $\beta = 1$ and which we denote by $S_\alpha^+$. Equation 5.4.6 of Uchaikin & Zolotarev (1999) states for $\alpha_0 \in (0, 1)$ and for all positive $\lambda$ one has: $\exp(-\lambda^{\alpha_0}) = \int_0^\infty \exp(-\lambda s)p_{S_{\alpha_0}^+}(s)ds$, where $p_{S_{\alpha_0}^+}(\cdot)$ is the density function of a positive $\alpha_0$-stable random variable, with no closed form in general. Using $\lambda = t^2, \alpha_0 = \alpha/2$ and the fact that $t^2 = |t|^2$, we obtain for $\alpha \in (0, 2)$, a Gaussian scale mixture representation over a random scale parameter $s$ for the symmetric $\alpha$-stable characteristic function as:

$$\exp(-|t|^\alpha) = \int_0^\infty \exp(-t^2 s)p_{S_{\alpha/2}^+}(s)ds. \tag{1}$$

This observation is exploited in Theorem 1 of Loría & Bhadra (2024), to represent the posterior of the stable process as a Gaussian process posterior, conditional on mixing positive $\alpha/2$-stable variables, which results in posterior sampling analogous to the GP case. However, in computing the covariance kernel of this GP, they suffer from the aforementioned combinatorial bottleneck.

Elliptical $\alpha$-stable random vectors (Samorodnitsky & Taqqu, 1994, Definition 2.5.1) centered at zero have a characteristic function defined similarly to the scalar case as: $\phi_{\mathbf{Z}}(\mathbf{t}) = \mathbb{E}[\exp(\mathrm{i}\mathbf{t}^T\mathbf{Z})] = \exp\{-(\mathbf{t}^T\Sigma\mathbf{t})^{\alpha/2}\}$, where $\Sigma$ is a positive definite matrix, termed the shape parameter. The main advantage of zero-centered elliptical $\alpha$-stable vectors is that they admit the Gaussian mixture representation: $\mathbf{Z} \stackrel{d}{=} S^{1/2}\mathbf{G}$, where $\stackrel{d}{=}$ denotes equality in distribution, $S \sim S_{\alpha/2}^+$ and $\mathbf{G} \sim \mathcal{N}(0, \Sigma)$, with $S, \mathbf{G}$ independent, which can be shown using a multivariate extension of Equation (1).

What the current work does is to find a recursive formula for deep kernel processes exploiting the technique of Cho & Saul (2009) to relate the conditional *stochastic covariance kernels* in a recursive manner, resulting in a far more computationally practical procedure in the kernel space rather than one in the feature space. One must note that although the limit processes are conditionally Gaussian for each layer, they possess stable margins. Nevertheless, the conditionally Gaussian representation naturally results in a stochastic covariance kernel in each layer, whose posterior can be learned given the data, to enable data-dependent representation learning. In this way, the proposed method bypasses the need for artificial noise injection in the manner of Aitchison et al. (2021).

## 1.2 SUMMARY OF MAIN CONTRIBUTIONS

1. The development of a novel deep $\alpha$-stable kernel process, arising as the infinite-width limit of a deep BNN under elliptical infinite variance priors on the weights at each layer. We further present a *conditionally Gaussian* representation of the resulting process at each layer, with a recursive formula relating these conditional covariance kernels, even when covariances for the marginal processes do not exist at any layer.

2. Theoretical and numerical demonstrations that the covariance kernel in the conditionally Gaussian representation for each layer is stochastic, allowing for learning a non-degenerate posterior of the kernels and identifying a clear demarcation with the deep GP literature.

3. A Markov chain Monte Carlo method for posterior sampling, which allows out-of-sample prediction at new inputs as well as uncertainty quantification of the predictions via the full posterior predictive distribution.

4. Numerical demonstrations in simulations and on benchmark UCI data sets that our method performs better in prediction than the competing methods, offers full predictive uncertainty estimates, and is considerably less time-consuming than the method of Loría & Bhadra (2024) that is limited to shallow (one hidden layer) BNNs.

## 2 Deep $\alpha$-stable kernel processes as infinite-width limits of deep Bayesian neural networks under heavy-tailed weights

Consider a deterministic input $\mathbf{x} \in \mathbb{R}^I$ with a corresponding one-dimensional output $\psi(\mathbf{x})$. We define an $L$ layer feedforward deep neural network (DNN) with $L-1$ hidden layers by the recursion:

$$f_j^{(\ell+1)}(\mathbf{x}) = g\left(b_j^{(\ell)} + \sum_{i=1}^{M_\ell} w_{ij}^{(\ell)} f_i^{(\ell)}(\mathbf{x})\right), \quad j = 1, \ldots, M_{\ell+1}; \ \ell = 1, \ldots, L-1, \tag{2}$$

$$\psi(\mathbf{x}) = \sum_{j=1}^{M_L} w_j^{(L)} f_j^{(L)}(\mathbf{x}), \tag{3}$$

where $f^{(1)} \equiv \mathbf{x}$, $g(\cdot)$ is a nonlinear activation function, and $M_\ell$ is the width of the $\ell$-th layer. Neal (1996) proved under mild conditions, with $L = 2$ and the weights $w_j^{(2)} \overset{ind.}{\sim} M_2^{-1/2} \mathcal{N}(0, 1)$, by taking the infinite-width limit $M_2 \to \infty$ for the last layer of this shallow network, the resulting stochastic process is a Gaussian process. Cho & Saul (2009) provided a recursive formula for relating the layer-wise covariance kernels when the activation function $g(\cdot)$ is of the form $g_\delta(\zeta) = \zeta^\delta \mathbf{1}_{\{\zeta > 0\}}$, for $\delta$ a non-negative integer, and for any number of hidden layers while taking the limits $M_1 \to \infty, \ldots, M_L \to \infty$. Unlike these works, we use *infinite variance* prior weights at each layer and study the resulting infinite-width limit posterior. For the first layer we take the weights $w_{mj}^{(1)} \overset{indep.}{\sim} \mathcal{N}(0, s_{m,+}^{(1)})$, where $s_{m,+}^{(1)} \overset{i.i.d.}{\sim} S_{\alpha/2}^+$, and the bias of the first layer is simply taken to be $b_j^{(1)} \overset{i.i.d.}{\sim} \mathcal{N}(0, 1)$. For the subsequent layers, we set $b_j^{(\ell)}$ as i.i.d. $\mathcal{N}(0, 1)$ and the weights $(w_{ij}^{(\ell)}$, for $i = 1, \ldots, M_\ell$, and $j = 1, \ldots, M_{\ell+1})$ of each layer $\ell > 1$ are given a distribution with infinite variance, as specified below in Theorem 1. This theorem establishes the resulting infinite-width limit as a non-Gaussian process with stable marginals. Further, similar to Tsuchida et al. (2019), it provides a *conditionally Gaussian* representation, conditioned on mixing positive $\alpha/2$ stable variables at each layer. The covariance kernel of this conditional GP can be derived explicitly under several commonly used activation functions, similarly to Cho & Saul (2009). Once this connection is established, posterior inference and prediction becomes feasible even in this infinite-variance regime.

**Theorem 1.** *For $j = 1, \ldots, M_{\ell+1}$ and $k = 1, \ldots, n$, define: $z_j^{(\ell)}(\mathbf{x}_k) = \frac{1}{M_\ell^{1/2}} \sum_{i=1}^{M_\ell} w_{ij}^{(\ell)} f_i^{(\ell)}(\mathbf{x}_k)$,*

*where $w_{ij}^{(\ell)} = (s_+^{(\ell)})^{1/2} \tilde{w}_{ij}^{(\ell)}$, with $s_+^{(\ell)}$ i.i.d. positive $\alpha/2$ stable random variables for $\ell = 2, \ldots, L$, and without loss of generality, $\tilde{w}_{ij}^{(\ell)}$ are i.i.d. with zero mean and unit variance and independent of $s_+^{(\ell)}$. Then, the weights $w_{ij}^{(\ell)}$ have infinite variance and as $M_\ell \to \infty$, the limiting distribution of $\mathbf{z}_j^{(\ell)} = (z_j^{(\ell)}(\mathbf{x}_1), \ldots, z_j^{(\ell)}(\mathbf{x}_n))$, in the $\ell$-th layer is elliptical $\alpha$-stable, with characteristic function:*

$$\phi_{\mathbf{z}_j^{(\ell)}|\Sigma^{(\ell)}}(\mathbf{t}) = \exp\left\{-(\mathbf{t}^T \Sigma^{(\ell)} \mathbf{t})^{\alpha/2}\right\},$$

*where $\mathbf{t} = (t_1, \ldots, t_n)$ and the $n \times n$ random matrix $\Sigma^{(\ell)}$ is positive definite with probability 1. As such, the limiting $\mathbf{z}_j^{(\ell)}$ admits the representation: $\mathbf{z}_j^{(\ell)} \mid s_+^{(\ell)}, \Sigma^{(\ell)} \sim \mathcal{N}(0, s_+^{(\ell)} \Sigma^{(\ell)})$, where*

$s_+^{(\ell)} \sim S_{\alpha/2}^+$. *For activation functions* $g_\delta(\zeta) = \zeta^\delta \mathbf{1}_{\{\zeta>0\}}$, *and* $\tilde{w}_{ij}^{(\ell)}$ *i.i.d. standard Gaussian, the matrix* $\Sigma^{(\ell)}$ *is recursively given as a function of* $\Sigma^{(\ell-1)}$ *and* $s_+^{(\ell-1)}$ *by:*

$$\Sigma_{k,h}^{(\ell)} = \pi^{-1} \left[ \left(1 + s_+^{(\ell-1)} \Sigma_{k,k}^{(\ell-1)}\right) \left(1 + s_+^{(\ell-1)} \Sigma_{h,h}^{(\ell-1)}\right) \right]^{\delta/2} J_\delta(\theta_{k,h}^{(\ell)}),$$

$$\theta_{k,h}^{(\ell)} = \cos^{-1} \left\{ \left[1 + s_+^{(\ell-1)} \Sigma_{k,h}^{(\ell-1)}\right] \left[1 + s_+^{(\ell-1)} \Sigma_{k,k}^{(\ell-1)}\right]^{-1/2} \left[1 + s_+^{(\ell-1)} \Sigma_{h,h}^{(\ell-1)}\right]^{-1/2} \right\},$$

*for* $k, h = 1, \ldots, n$, $\ell = 2, \ldots, L$; *where* $J_\delta(\theta)$ *can be computed explicitly for* $\delta \in \mathbb{N}$ *using Eq. (4) of* Cho & Saul (2009), *and* $\Sigma^{(1)} = \mathbf{X}\mathbf{S}_+^{(1)}\mathbf{X}^T$, *where* $\mathbf{X} = [\mathbf{x}_1, \ldots, \mathbf{x}_n]^T$, $\mathbf{S}_+^{(1)} = \text{diag}(s_{1,+}^{(1)}, \ldots, s_{I,+}^{(1)})$, *with* $s_{m,+}^{(1)} \overset{i.i.d.}{\sim} S_{\alpha/2}^+$ *for* $m = 1, \ldots, I$.

A proof is in Appendix B. For completeness, we note that $g_0(\zeta)$ and $g_1(\zeta)$ correspond to respectively the step function and ReLU activations, for which $J_0(\theta) = \pi - \theta$ and $J_1(\theta) = \sin(\theta) + (\pi - \theta)\cos(\theta)$, as derived by Cho & Saul (2009). This result has similarities to those previously obtained by both Lee et al. (2018) and de G. Matthews et al. (2018). However, we emphasize the key difference that in our case the kernels $s_+^{(\ell)}\Sigma^{(\ell)}$ at all layers are *random* for $\alpha < 2$, conditional on scales with a positive $\alpha/2$-stable distribution, although they are positive definite with probability one. The limiting kernel for the Gaussian ($\alpha \to 2$) case reduces to a deterministic one, as in earlier works, since $S_{\alpha/2 \to 1}^+$ converges to a degenerate point mass at 1 with probability one. This permits an explicit characterization of a deep kernel process with marginally infinite variance at each layer via a conditional deep Gaussian process. Two significant gains, when compared to Loría & Bhadra (2024) are obtained: (1) there is no need for exponential complexity computations in the kernel space as opposed to in the feature space, and (2) our derivations work with multi-layer BNNs, in contrast to the results of Loría & Bhadra (2024) that are limited to BNNs with a *single* hidden layer.

The next proposition describes the posterior predictive density implied by the probabilistic structure in Theorem 1. The posterior predictive density we present combines the simplicity of predicting in GPs while ameliorating the challenges posed by variables that have infinite variance.

**Proposition 2.** *Consider a vector* $\mathbf{y}$ *of* $n$ *real-valued observations with corresponding inputs* $\mathbf{X} = [\mathbf{x}_1, \ldots, \mathbf{x}_n]^T$, *where* $\mathbf{x}_k \in \mathbb{R}^I$ *for all* $n$. *Consider the model of Equations (2) and (3) and suppose we observe data with an additive error:* $y_k = \psi(\mathbf{x}_k) + \varepsilon_k$ *where* $\varepsilon_k \overset{i.i.d.}{\sim} \mathcal{N}(0, \sigma^2)$. *Then, under the setting of Theorem 1, and taking* $M_\ell \to \infty$ *for all* $\ell$, *at* $m$ *new inputs* $\mathbf{X}^* = [\mathbf{x}_{n+1}, \ldots, \mathbf{x}_{n+m}]^T$, *the posterior predictive distribution of the observations* $\mathbf{y}^*$ *is given by:*

$$\mathbf{y}^* \mid \mathbf{y}, \mathbf{X}, \mathbf{X}^*, \{s_+^{(\ell)}\}_{\ell=2}^L, \mathbf{S}_+^{(1)} \sim \mathcal{N}_m(\boldsymbol{\mu}^*, \boldsymbol{\Lambda}^*), \text{ where,}$$

$$\boldsymbol{\mu}^* = \boldsymbol{\Lambda}_{(n+1):(n+m),1:n} \boldsymbol{\Lambda}_{1:n,1:n}^{-1} \mathbf{y},$$

$$\boldsymbol{\Lambda}^* = \boldsymbol{\Lambda}_{(n+1):(n+m),(n+1):(n+m)} - \boldsymbol{\Lambda}_{(n+1):(n+m),1:n} \boldsymbol{\Lambda}_{1:n,1:n}^{-1} \boldsymbol{\Lambda}_{1:n,(n+1):(n+m)},$$

$$s_+^{(\ell)} \overset{i.i.d.}{\sim} S_{\alpha/2}^+; s_{m,+}^{(1)} \overset{i.i.d.}{\sim} S_{\alpha/2}^+, \text{ for, } \ell = 2, \ldots, L; \ m = 1, \ldots, I,$$

*and* $\boldsymbol{\Lambda} = \Sigma^{(L)} + \sigma^2 \mathbf{I}_{n+m}$, *where* $\Sigma^{(L)}$ *is the matrix obtained in the last layer in Theorem 1 with the inputs, in order,* $[\mathbf{X}, \mathbf{X}^*]$ *and* $\mathbf{I}_{n+m}$ *denotes the identity matrix of size* $(n+m)$.

Proof of Proposition 2 is in Appendix C. This proposition enables computation of the kernel posterior and prediction, and is key to the numerical demonstrations presented later. We also remark that the purpose of Proposition 2 is to tackle regression tasks rather than classification. However, similar results can also be obtained for classification. Note that for $\alpha < 2$, the conditional kernel $\boldsymbol{\Lambda}^*$ is a stochastic matrix, which we verify numerically in Sections 3 and 4.

Making use of Proposition 2, the next algorithm provides a Markov chain Monte Carlo (MCMC) procedure for sampling from the posterior predictive density $p(\mathbf{y}^*, \{s_+^{(\ell)}\}_{\ell=2}^L, \mathbf{S}_+^{(1)} \mid \mathbf{y}, \mathbf{X}, \mathbf{X}^*)$. Algorithm 1 obtains a valid sample of the posterior density $p(\mathbf{y}^* \mid \mathbf{y}, \mathbf{X}, \mathbf{X}^*)$ by classic results in the MCMC literature (Chib & Greenberg, 1995). Our algorithm also provides valid samples of the posterior stochastic covariance matrix induced by the kernel at the design points. We remark that simulating in previously unseen locations can be done in an off-line manner, by simply utilizing the scales and the collection of input points, which permits an efficient prediction algorithm without

---

**Algorithm 1** A Metropolis–Hastings sampler for the posterior predictive distribution of the deep $\alpha$-kernel process.

---

**Require:** Observations $\mathbf{y} \in \mathbb{R}^n$, with $I$-dimensional input variables $\mathbf{X} \in \mathbb{R}^{n \times I}$, new input variables $\mathbf{X}^* \in \mathbb{R}^{m \times I}$, and number of MCMC iterations $T$.

**Output:** Posterior predictive samples $\{\mathbf{y}_t^*\}_{t=1}^T$, samples of stochastic matrix $\{\Sigma_t^{(L)}\}_{t=1}^T$.

    Simulate starting scales $\mathbf{S}_{+,(0)}^{(1)}, s_{+,(0)}^{(\ell)} \overset{i.i.d.}{\sim} S_{\alpha/2}^+$ for $\ell = 2, \ldots, L$ using the algorithm of Chambers et al. (1976).

    **for** $t = 1, \ldots, T$ **do**

        Simulate $\mathbf{S}_{+,(t)}^{(1)} \mid \{s_{+,(t-1)}^{(\ell)}\}_{\ell=2}^L$ using Algorithm 2.

        **for** $\ell = 2, \ldots, L-1$ **do**

            Simulate $s_{+,(t)}^{(\ell)} \mid \mathbf{y}, \{s_{+,(t-1)}^{(h)}\}_{h=\ell+1}^L$ using Algorithm 2.

        **end for**

        Simulate $s_{+,(t)}^{(L)} \mid \mathbf{y}$ using Algorithm 3.

        Compute $\boldsymbol{\mu}_t^*$ and $\boldsymbol{\Lambda}_t^*$ using $\Sigma_t^{(L)}$ in Proposition 2.

        Simulate $\mathbf{y}_t^* \mid (\mathbf{y}, \Sigma_t^{(L)}) \sim \mathcal{N}_m(\boldsymbol{\mu}_t^*, \boldsymbol{\Lambda}_t^*)$.

    **end for**

    **return** $\{\mathbf{y}_t^*\}_{t=1}^T$ and $\{\boldsymbol{\Lambda}_t\}_{t=1}^T$.

---

a need to resample the scales. We also make use of a half-Cauchy prior (Gelman, 2006) on the variance of the errors $\sigma^2$, to specify a fully-Bayesian model. A full implementation, with examples, is freely available at: `https://github.com/loriaJ/deep-alpha-kernel`.

Following the construction of Theorem 1 ensures we have defined a valid kernel process that is consistent under marginalization, in the manner described by Aitchison et al. (2021). Specifically, the process $\{\Sigma^{(\ell)}\}_{\ell=1}^L$ is a deep kernel process with probability 1, meaning that they induce a distribution on positive definite matrices of different sizes which are consistent under marginalization. The proof of this follows from a similar argument as in Aitchison et al. (2021), and reduces to their result under fixed scales. For completeness we provide further elaboration in Appendix E.

Additionally, the process $\{\Sigma^{(\ell)}\}_{\ell=1}^L$ has an implicit dependence on $\alpha$, the index of the $\alpha$-stable random variables. For $\alpha = 2$, the characteristic function of $\mathbf{z}_j^{(\ell)}$ becomes that of a Gaussian random variable, and $\Sigma^{(\ell)}$ is deterministic. Remarkably this kernel process is stochastic when $\alpha < 2$. The importance of having a stochastic deep kernel process is that it allows us to learn features, which deep Gaussian processes are not able to do, as highlighted by Yang et al. (2023). We formalize this notion in the following proposition, with proof in Appendix F.

**Proposition 3.** *For $\alpha < 2$ the posterior distribution of the features $\mathbf{z}_j^{(\ell)}$ is given by*

$$p(\mathbf{z}_j^{(\ell)} \mid \mathbf{X}, \mathbf{y}) = \int p(\mathbf{z}_j^{(\ell)} \mid \{s_+^{(\ell)}\}_{\ell=2}^L, \mathbf{S}_+^{(1)}) p(\{s_+^{(\ell)}\}_{\ell=2}^L, \mathbf{S}_+^{(1)} \mid \mathbf{X}, \mathbf{y}) \prod_{\ell=2}^L ds_+^{(\ell)} \prod_{m=1}^I ds_{m,+}^{(1)}.$$

*This implies that the features also depend on the observations. For $\alpha = 2$, the posterior distribution of the features are independent of the observations, and hence, cannot be learned from the data.*

The importance of this proposition is that we are able to verify that in the deep kernel process we have defined, the features *do* depend on the observations. As such, there is feature learning only when $\alpha < 2$, but when $\alpha = 2$ we return to a deterministic kernel and are not able to learn the features. As an additional benefit, the proposition provides a straightforward procedure for sampling from the posterior of the features. Namely, we can sample the scales from $p(\{s_+^{(\ell)}\}_{\ell=2}^L, \mathbf{S}_+^{(1)} \mid \mathbf{X}, \mathbf{y})$, and then sample from $\mathbf{z}_j^{(\ell)} \mid \{s_+^{(\ell)}\}_{\ell=2}^L, \mathbf{S}_+^{(1)} \sim \mathcal{N}(0, s_+^{(\ell)} \Sigma^{(\ell)})$ using the posterior samples of $s_+^{(\ell)}$.

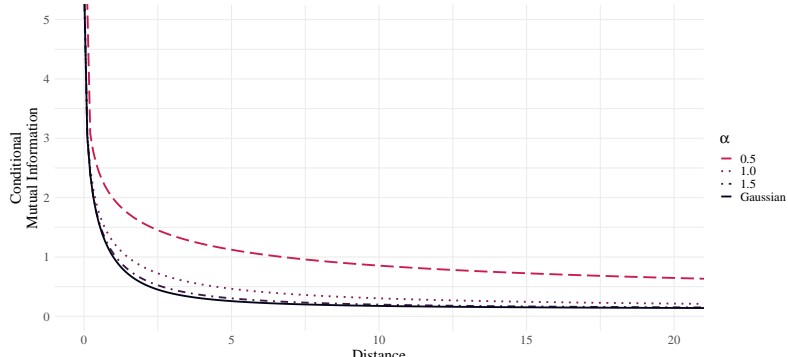

Figure 1: Decay of the conditional mutual information for the deep $\alpha$-kernel process as a function of the distance between the inputs with $L = 2, \delta = 1$. The limiting Gaussian case ($\alpha = 2$) is also included.

## 3 NUMERICAL EXPERIMENTS

### 3.1 CONDITIONAL MUTUAL INFORMATION AS A FUNCTION OF $\alpha$

Figure 1 displays the conditional mutual information for different values of $\alpha$ at varying distances over a uniform grid in one-dimension with $L = 2, \delta = 1$, for the proposed deep $\alpha$-kernel process, i.e., a process with one hidden layer. For a GP, the mutual information is simply a function of the correlation, but this quantity is not well defined in our case, since there is no well-defined covariance. As such, we need to refer to the conditional mutual information (Cover & Thomas, 2006, p. 23), which can be computed using the conditionally-Gaussian representation and numerical integration via Monte Carlo methods. The expressions of conditional mutual information and details of computation are provided in Appendix G. The key finding is that the conditional mutual information decays at a slower rate for smaller $\alpha$. The long memory behavior of the conditional mutual information suggests the developed deep $\alpha$-kernel processes to be especially adept at picking up distant relationships, which would be modeled as nearly independent under a GP.

### 3.2 POSTERIOR PREDICTION AND UNCERTAINTY QUANTIFICATION UNDER DISCONTINUITY

We display results on functions with jumps in one, two, and ten dimensions. The reason we focus specifically on discontinuous true functions is that they are misspecified under a GP prior under *any* covariance function, which can model varying levels of smoothness, but not a lack of mean square continuity. In contrast, heavy-tailed priors enable posterior inference on a broader function class, e.g., those belonging to Besov space; see recent theoretical works by Agapiou & Castillo (2024) and extensive numerical works in Loría & Bhadra (2024). The comparisons include a Bayesian GP method (Gramacy & Taddy, 2010) and a frequentist approach on GPs (Dancik & Dorman, 2008), as well as the DIWP and NNGP method as implemented by Aitchison et al. (2021), and the Stable method by Loría & Bhadra (2024). We refer to our proposed method as D$\alpha$-KP as a shorthand for deep $\alpha$-stable kernel process. For the deep kernel methods of Aitchison et al. (2021) we train all the models using the same hyperparameters they use, with 8000 total steps with $10^{-2}$ as the step-size for the first half, and $10^{-3}$ for the second half. For the D$\alpha$-KP method we run 3000 MCMC simulations with $\alpha = 1, \delta = 1$. For a fair comparison, the three kernel methods all use $L = 2$, meaning they are shallow with one hidden layer. The following simulation settings are considered, respectively in one, two, and ten dimensions, all under discontinuous truth.

1. The true function in one dimension is $f(\xi) = 5 \times \mathbf{1}_{\{\xi > 0\}}$ and we generate observations as $y(\xi) = f(\xi) + \varepsilon; \ \varepsilon \sim \mathcal{N}(0, 0.5^2)$. For training we consider 40 equally-spaced input points on $[-1, 1]$ and predict on 100 out-of-sample points.

2. In two dimensions we use the function $f(\xi_1, \xi_2) = 5 \times \mathbf{1}_{\{\xi_1 > 0\}} + 5 \times \mathbf{1}_{\{\xi_1 > 0\}}$ and generate $y(\xi_1, \xi_2) = f(\xi_1, \xi_2) + \varepsilon; \ \varepsilon \sim \mathcal{N}(0, 0.5^2)$, using a $7 \times 7$ uniform grid on $[-1, 1]^2$ for training, and a similar $9 \times 9$ grid for testing.

3. In the ten-dimensional setting we use the function $f(\boldsymbol{\xi}) = 6\text{sign}(\xi_1) + 8\text{sign}(\xi_2 + \xi_3) + 6\text{sign}(\xi_4 + \xi_5) + 6\text{sign}(\xi_6 + \xi_7) + 6\text{sign}(\xi_8 + \xi_9) + 6\text{sign}(\xi_{10})$ and generate $y(\boldsymbol{\xi}) = f(\boldsymbol{\xi}) + \varepsilon$, with $\varepsilon \sim \mathcal{N}(0, 0.5^2)$. The design points are generated by $\xi_i \overset{i.i.d.}{\sim} \text{Unif}(-0.5, 0.5)$. We generate 20 splits, each with 300 training and 300 testing observations.

Under these simulation settings, we provide in Table 1 the prediction errors for the competing methods by presenting the root mean squared error (RMSE) and mean absolute error (MAE). Our method obtains smaller out-of-sample prediction errors compared to the other methods, and is generally a close second to the Stable method of Loría & Bhadra (2024). However, the running time of our method is a fraction of the time of Loría & Bhadra (2024), which we highlight in Supplementary Section H.1. As a result of this high run time owing to the exponential complexity computations, the results for the Stable method are only available in one and two dimensions. Figure 2a compares the function fit for the one-dimensional function. Figure 2b displays the uncertainty quantification (UQ) by presenting 90% posterior predictive intervals, where the benefits of the methods using $\alpha$-stable priors over GP priors are clear under discontinuity, as is the inability of GP to model such functions. We display similar figures for the examples in two and ten dimensions in Appendix H. Furthermore, the out-of-sample prediction errors for this case demonstrate the superior performance of methods that incorporate $\alpha$-stable random variables. Appendix H.2 shows the percent of observations that are covered in the 90% posterior predictive intervals for the simulation settings under all methods. In Appendix H.3 we present additional results, including an ablation study on the parameter $\alpha$ (Appendix H.3.3) and visualizations of the posterior quantiles of the random kernel matrix (Appendix H.3.4).

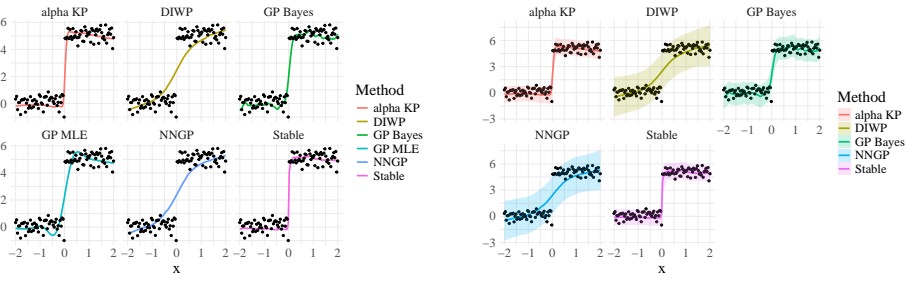

(a) Function fit for the different methods.

(b) 90% posterior predictive intervals for the Bayesian methods.

Figure 2: Function fit and uncertainty quantification for the competing methods for a 1-d function with a single jump.

Table 1: Out-of-sample errors in numerical examples, in twenty different splits. Best in **bold**. Stable not available for more than 2 dimensions.

| Method | One Dimension | | Two Dimensions | | Ten Dimensions | |
|---|---|---|---|---|---|---|
| | RMSE (SD) | MAE (SD) | RMSE (SD) | MAE (SD) | RMSE (SD) | MAE (SD) |
| D$\alpha$-KP | 0.57 (0.05) | 0.45 (0.04) | 0.86 (0.09) | 0.67 (0.07) | **8.08** (0.38) | **6.48** (0.33) |
| DIWP | 1.08 (0.04) | 0.84 (0.03) | 1.69 (0.05) | 1.36 (0.05) | 8.92 (0.38) | 7.21 (0.32) |
| GP Bayes | 0.69 (0.05) | 0.52 (0.04) | 0.90 (0.06) | 0.70 (0.06) | 10.39 (0.63) | 8.32 (0.52) |
| GP MLE | 0.77 (0.06) | 0.57 (0.04) | 1.19 (0.08) | 0.92 (0.07) | 8.32 (0.38) | 6.68 (0.34) |
| NNGP | 1.08 (0.04) | 0.84 (0.03) | 1.69 (0.05) | 1.36 (0.04) | 8.92 (0.39) | 7.21 (0.34) |
| Stable | **0.52** (0.03) | **0.42** (0.03) | **0.57** (0.08) | **0.45** (0.04) | – | – |

### 3.3 THE EFFECT OF DEPTH ON PREDICTION FOR $\alpha$-KERNEL PROCESSES

We numerically investigate the effect of depth on predictive performance in the synthetic data sets of Section 3.2, with $L = 2$ results given there. As can be seen from Table 2, there is no meaningful difference between deep and shallow $\alpha$ kernel processes for these simulation settings. Recent works

have theoretically investigated fundamental barriers to *trainable depth* of deep neural networks, exceeding which actually results in degraded performance (Schoenholz et al., 2017). A similar theoretical investigation is beyond the scope of the current work, but we conjecture that the resultant stable process can capture a function class *rich enough* with even one hidden layer, so that the effect of depth becomes a secondary issue for these test functions. This is a very different situation from a deep NNGP, which can only capture functions that are mean square continuous, with the effect of depth primarily manifesting in a richer covariance kernel, but not in a non-Gaussian process.

Table 2: Average (SD) out-of-sample errors for deep $\alpha$-KP, with $\delta = 1, \alpha = 1$, in the three simulation scenarios with varying layer depth, over twenty different splits.

| | One Dimension | | Two Dimensions | | Ten Dimensions | |
|---|---|---|---|---|---|---|
| L | RMSE (SD) | MAE (SD) | RMSE (SD) | MAE (SD) | RMSE (SD) | MAE (SD) |
| 3 | 0.56 (0.05) | 0.45 (0.04) | 0.82 (0.07) | 0.65 (0.05) | 8.09 (0.38) | 6.49 (0.34) |
| 6 | 0.57 (0.05) | 0.45 (0.04) | 0.81 (0.07) | 0.64 (0.05) | 8.10 (0.39) | 6.49 (0.34) |
| 11 | 0.58 (0.05) | 0.45 (0.04) | 0.85 (0.06) | 0.66 (0.05) | 8.11 (0.40) | 6.51 (0.35) |
| 16 | 0.58 (0.05) | 0.46 (0.04) | 0.88 (0.06) | 0.69 (0.05) | 8.11 (0.39) | 6.50 (0.34) |

## 4 APPLICATIONS TO UCI DATA SETS

### 4.1 OUT-OF-SAMPLE PREDICTIVE PERFORMANCE

We apply the five methods (D$\alpha$-KP, DIWP, NNGP, GP Bayes, and GP MLE) to the three well-known data sets from the UCI repository: Boston, Yacht, and Energy. Due to the size of these data sets and the $\mathcal{O}(n^{I+2})$ computational complexity, the Stable procedure was not feasible to run. To this end, we split each of the data sets in 20 different folds, training in 19 and testing in the remaining fold; and repeat the process for each of the folds. We display the average RMSE and average MAE in Table 3, with the respective standard deviations. We find that in two out of three cases (Energy and Yacht), D$\alpha$-KP has the best predictive performance, while being a close second on the Boston data set. In Appendix I.1 we display the running times for the considered methods, and in Appendix I.2 we display the coverage of the credible intervals.

Table 3: Comparison of out-of-sample errors in twenty splits of the three UCI datasets, using the D$\alpha$-KP ($\alpha = 1, \delta = 1, L = 2$), DIWP, NNGP (Aitchison et al., 2021), GP Bayes, and GP MLE. Number of observations denoted by $n$ and number of inputs by $I$. Stable method not available for $I > 2$. Best in **bold**.

| | Boston ($n = 506, I = 13$) | | Energy ($n = 769, I = 8$) | | Yacht ($n = 308, I = 6$) | |
|---|---|---|---|---|---|---|
| Method | RMSE (SD) | MAE (SD) | RMSE (SD) | MAE (SD) | RMSE (SD) | MAE (SD) |
| D$\alpha$-KP | 2.59 (0.73) | 1.78 (0.35) | **0.46** (0.07) | **0.32** (0.05) | **0.31** (0.12) | **0.16** (0.05) |
| DIWP | 2.85 (0.89) | 2.01 (0.41) | 0.48 (0.06) | 0.34 (0.04) | 0.60 (0.20) | 0.30 (0.10) |
| NNGP | 3.00 (0.87) | 2.04 (0.43) | 2.18 (0.23) | 1.57 (0.18) | 3.88 (0.87) | 2.58 (0.49) |
| GP Bayes | **2.58** (0.75) | **1.76** (0.35) | 0.68 (0.06) | 0.51 (0.04) | 0.48 (0.23) | 0.22 (0.07) |
| GP MLE | 3.93 (1.02) | 2.58 (0.51) | 0.49 (0.06) | 0.34 (0.04) | 0.52 (0.34) | 0.25 (0.11) |
| Stable | – | – | – | – | – | – |

### 4.2 NON-GAUSSIAN FEATURE LEARNING

In this section we numerically investigate whether D$\alpha$-KP indeed learns a non-degenerate and non-Gaussian posterior of the features. Specifically, we display in Figure 3 the first two features in the hidden layer of a two layer D$\alpha$-KP for the Energy data set. The posterior of features in the $\ell$-th layer is given by: $\mathbf{z}_j^{(\ell)} \mid \Sigma^{(\ell)}, \mathbf{y} \sim \mathcal{N}(0, \Sigma^{(\ell)})$. Note that after marginalizing over $\Sigma^{(\ell)}$, which has stable dependence, Proposition 3 indicates we would obtain a non-Gaussian distribution. Figure 3 confirms the non-Gaussianity. In this figure we include the 2-d heat map of posterior samples with

1-d marginal box plots, which show far more values in the tails than what can be expected under Gaussianity. The normal q–q plots also confirm the heavy tails. We include similar figures for the Boston and Yacht datasets in Appendix J. The non-Gaussianity of $\mathbf{z}_j^{(\ell)}$ also serves as confirmation that $\Sigma^{(\ell)}$ is indeed stochastic, since otherwise we would obtain Gaussian features.

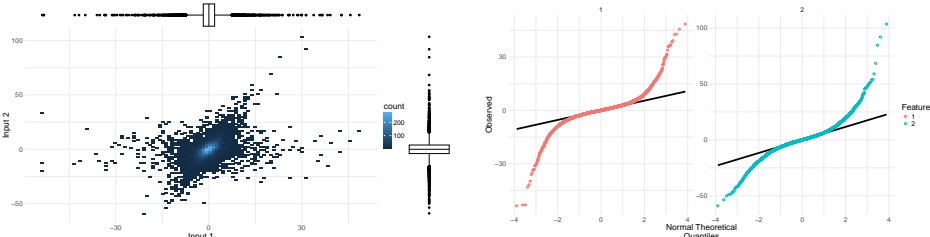

(a) Scatterplot of the first two features, with box plots for the marginals.

(b) Normal q–q plot of the first two features.

Figure 3: Posterior distribution of the features for the Energy data set using 10k posterior MCMC samples with 1k burn-in samples.

## 5 CONCLUDING REMARKS

We present a viable posterior inference procedure for deep $\alpha$-stable kernel processes arising as the infinite-width limit of a BNN under infinite variance prior weights at each layer. Through a convenient representation of elliptical $\alpha$-stable random vectors as a Gaussian mixture with respect to positive $\alpha/2$ stable variables acting as random scales, we obtain an explicit relationship with a deep conditionally Gaussian process, upon which the result of Cho & Saul (2009) can be leveraged to link the conditional covariance kernels in a recursive manner, resulting in a computationally viable method. We also present extensive evidence of benefits in prediction and uncertainty quantification under a true data generating function that contains jump-type discontinuities. Compared to the approach of Loría & Bhadra (2024), the current work overcomes the serious limitations of being restricted to one hidden layer and exponential complexity computations.

Several future directions could naturally follow from the current work. Although MCMC is gold standard for full posterior uncertainty quantification, for computational scalability, one may look for a variational approximation to the posterior predictive density we derived. Applying techniques such as inducing points (Snelson & Ghahramani, 2005), would also help scale the algorithm to data sets with larger sample sizes.

Our results indicate successful feature learning under non-Gaussian stochastic processes that is not possible under a GP (shallow or deep) and achieves this without any artificial noise injection in the manner of Aitchison et al. (2021). In the current work, we have not made any efforts of feature selection, which appears to be a promising direction under the infinite-width limits of suitable sparsity inducing architectures, such as dropout (Srivastava et al., 2014), that should now be possible to study following our work and leveraging its main ideas. More general activation functions, as in Tsuchida et al. (2021), may also be considered.

Although in this paper we consider the infinite-width limit of deep Bayesian neural networks, another interesting GP limit arises due to the dynamics of the stochastic gradient descent (SGD) noise during the training of the network, resulting in the neural tangent kernel (NTK) (Jacot et al., 2018), under the assumption of *bounded variance SGD noise*. However, recent works, including Simsekli et al. (2019), have found empirical evidence that SGD noise can be heavy-tailed. In this regime, one should expect a non-Gaussian analog for the NTK, with a *random* kernel, whose data-dependent posterior should be possible to study using techniques similar to ours.

ACKNOWLEDGMENTS

Bhadra was supported by U.S. National Science Foundation Grant DMS-2014371.

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

## A  THE GENERALIZED CENTRAL LIMIT THEOREM

**Theorem 4.** *(Uchaikin & Zolotarev, 1999, p. 62) Let $X_1, \ldots, X_n$ be independent and identically distributed random variables with the distribution function $F_X(x)$ satisfying the conditions:*

$$1 - F_X(x) \sim cx^{-\mu}, x \to \infty,$$

$$F_X(x) \sim dx^{-\mu}, x \to -\infty,$$

*with $\mu > 0$, $c \geq 0, d \geq 0$. Then, there exists sequences $a_n$ and $b_n$ such that the distribution of the centered and normalized sum $Z_n = b_n^{-1}(\sum_{i=1}^n X_i - a_n)$, converges in distribution to the stable distribution with parameters:*

$$\alpha = \min(\mu, 2), \ \beta = \frac{c-d}{c+d},$$

*as $n \to \infty$, i.e., $F_{Z_n}(x) \to G(x; \alpha, \beta)$ for all $x$ where $G$ is continuous, where $F_{Z_n}$ is the c.d.f of $Z_n$, and $G$ is the c.d.f. of an $\alpha$-stable random variable with symmetry parameter $\beta$. The coefficients $a_n$ and $b_n$ are given in Table 4.*

Table 4: Centering and normalizing coefficients $a_n$ and $b_n$ for the generalized central limit theorem.

| $\mu$ | $\alpha$ | $a_n$ | $b_n$ |
|---|---|---|---|
| $0 < \mu < 1$ | $\mu$ | $0$ | $\pi^{1/\alpha}(c+d)^{1/\alpha}[2\Gamma(\alpha)\sin(\alpha\pi/2)]^{-1/\alpha}n^{1/\alpha}$ |
| $\mu = 1$ | $\mu$ | $(c-d)n\log(n)$ | $\pi(c+d)n/2$ |
| $1 < \mu < 2$ | $\mu$ | $n\mathbb{E}[X]$ | $(c+d)^{1/\alpha}[2\Gamma(\alpha)\sin(\alpha\pi/2)]^{-1/\alpha}n^{1/\alpha}$ |
| $\mu = 2$ | $2$ | $n\mathbb{E}[X]$ | $(c+d)^{1/2}[n\log(n)]^{1/2}$ |
| $\mu > 2$ | $2$ | $n\mathbb{E}[X]$ | $[(1/2)\mathrm{Var}(X)]^{1/2}n^{1/2}$ |

## B  PROOF OF THEOREM 1

First, the marginal variance of $w_{ij}^{(\ell)}$ is unbounded:

$$\mathbb{V}(w_{ij}^{(\ell)}) = \mathbb{V}[(s_+^{(\ell)})^{1/2}\mathbb{E}[\tilde{w}_{ij}^{(\ell)} \mid s_+^{(\ell)}]] + \mathbb{E}[s_+^{(\ell)}\mathbb{V}[\tilde{w}_{ij}^{(\ell)} \mid s_+^{(\ell)}]] = \infty,$$

since for Stable $(\alpha/2)$ random variables, the expectation is unbounded. Next, we have:

$$z_j^{(\ell)}(\mathbf{x}_k) = \frac{1}{M_\ell^{1/2}}\sum_{i=1}^{M_\ell} w_{ij}^{(\ell)} f_i^{(\ell)}(\mathbf{x}_k) = (s_+^{(\ell)})^{1/2}\frac{1}{M_\ell^{1/2}}\sum_{i=1}^{M_\ell} \tilde{w}_{ij}^{(\ell)} f_i^{(\ell)}(\mathbf{x}_k).$$

Since $\tilde{w}_{ij}^{(\ell)}$ are i.i.d. with unit variance by assumption, we have, as $M_\ell \to \infty$:

$$\frac{1}{M_\ell^{1/2}}\sum_{i=1}^{M_\ell} \tilde{w}_{ij}^{(\ell)} f_i^{(\ell)}(\mathbf{x}_k) \overset{D}{\to} \mathcal{N}(0, \lambda_k^{(\ell)}),$$

by an application of the classical central limit theorem, where $\overset{D}{\to}$ denotes convergence in distribution. Thus,

$$z_j^{(\ell)}(\mathbf{x}_k) \overset{D}{\to} (s_+^{(\ell)})^{1/2}\mathcal{N}(0, \lambda_k^{(\ell)}),$$

which is an $\alpha$-stable random variable for each $j, k$, and $\lambda_k^{(\ell)} = \mathbb{V}(f_i^{(\ell)}(\mathbf{x}_k) \mid \{s_+^{(l)}\}_{l=1}^{\ell-1}, \mathbf{S}_+^{(1)})$, which does not depend on $i$ since the $f_i^{(\ell)}(\mathbf{x}_k)$ are i.i.d. across $i$.

The conditional covariance between two features in layer one is given by:

$$\mathrm{Cov}(z_j^{(1)}(\mathbf{x}_k), z_j^{(1)}(\mathbf{x}_{k'}) \mid \mathbf{S}_+^{(1)}) = \mathrm{Cov}\left(\sum_{m=1}^I (s_{m,+}^{(1)})^{1/2}\tilde{w}_{mj}^{(1)} x_{m,k}, \sum_{m'=1}^I (s_{m',+}^{(1)})^{1/2}\tilde{w}_{m'j}^{(1)} x_{m',k'} \mid \mathbf{S}_+^{(1)}\right)$$

$$= \sum_{m'=1}^I (s_{m',+}^{(1)})^{1/2}x_{m',k'}\sum_{m=1}^I (s_{m,+}^{(1)})^{1/2}x_{m,k}\mathrm{Cov}(\tilde{w}_{ij}^{(1)}, \tilde{w}_{m'j}^{(1)} \mid \mathbf{S}_+^{(1)})$$

$$= \sum_{m=1}^I s_{m,+}^{(1)}x_{m,k}x_{m,k'}.$$

As such the conditional covariance matrix of $\mathbf{z}_j^{(1)} \mid \mathbf{S}_+^{(1)}$ is $\Sigma^{(1)} = \mathbf{X}\mathbf{S}_+^{(1)}\mathbf{X}^T$. Now that we have the first covariance matrix, we proceed by induction over $\ell$, with the note that in the case $\ell = 2$ what we denote by $s_+^{(1)}$ corresponds to $\mathbf{S}_+^{(1)}$. Next we obtain the characteristic function of $\mathbf{z}_j^{(\ell)}$ for a finite $M_\ell$, assuming that $\Sigma^{(\ell-1)}, s_+^{(\ell)}, s_+^{(\ell-1)}$ are given. To this end, for $\mathbf{t} = (t_1, \ldots, t_n) \in \mathbb{R}^n$:

$$
\begin{aligned}
\phi_{\mathbf{z}_j^{(\ell)} \mid \Sigma^{(\ell-1)}, s_+^{(\ell)}, s_+^{(\ell-1)}}(\mathbf{t}) &= \mathbb{E}\left[\exp\left\{i\mathbf{t}^T \mathbf{z}_j^{(\ell)}\right\} \mid \Sigma^{(\ell-1)}, s_+^{(\ell)}, s_+^{(\ell-1)}\right] \\
&= \mathbb{E}\left[\exp\left\{i\sum_{k=1}^n \sum_{i=1}^{M_\ell} t_k M_\ell^{-1/2} w_{ij}^{(\ell)} f_i^{(\ell)}(\mathbf{x}_k)\right\} \mid \Sigma^{(\ell-1)}, s_+^{(\ell)}, s_+^{(\ell-1)}\right] \\
&= \mathbb{E}\left[\exp\left\{i\sum_{k=1}^n \sum_{i=1}^{M_\ell} t_k M_\ell^{-1/2} (s_+^{(\ell)})^{1/2} \tilde{w}_{ij}^{(\ell)} f_i^{(\ell)}(\mathbf{x}_k)\right\} \mid \Sigma^{(\ell-1)}, s_+^{(\ell)}, s_+^{(\ell-1)}\right] \\
&= \mathbb{E}\left[\exp\left\{i(s_+^{(\ell)})^{1/2} M_\ell^{-1/2} \sum_{i=1}^{M_\ell} \tilde{w}_{ij}^{(\ell)} \sum_{k=1}^n t_k f_i^{(\ell)}(\mathbf{x}_k)\right\} \mid \Sigma^{(\ell-1)}, s_+^{(\ell)}, s_+^{(\ell-1)}\right] \\
&= \mathbb{E}\left[\exp\left\{i(s_+^{(\ell)})^{1/2} M_\ell^{-1/2} \sum_{i=1}^{M_\ell} \tilde{w}_{ij}^{(\ell)} A_i^{(\ell)}\right\} \mid \Sigma^{(\ell-1)}, s_+^{(\ell)}, s_+^{(\ell-1)}\right], \quad (4)
\end{aligned}
$$

where we define $A_i^{(\ell)} = \sum_{k=1}^n t_k f_i^{(\ell)}(\mathbf{x}_k)$, which are independent and identically distributed over $i$, and the expectation $\mathbb{E}(\cdot)$ is over the distribution of $\tilde{w}_{ij}^{(\ell)}$. Next, the expectation of $\tilde{w}_{ij}^{(\ell)} A_i^{(\ell)}$ is zero since both factors of the product are independent and the mean of one of them is zero, and the variance is $\mathbb{E}[(\sum_{k=1}^n t_k f_i^{(\ell)}(\mathbf{x}_k))^2 \mid \Sigma^{(\ell-1)}, s_+^{(\ell-1)}]$.

Taking the limit $M_\ell \to \infty$ of the expression inside the exponential in Equation (4), we obtain:

$$
\lim_{M_\ell \to \infty} (s_+^{(\ell)})^{1/2} M_\ell^{-1/2} \sum_{i=1}^{M_\ell} \tilde{w}_{ij}^{(\ell)} A_i^{(\ell)} \xrightarrow{D} (s_+^{(\ell)})^{1/2} \mathcal{N}\left(0, \mathbb{E}\left[\left(\sum_{k=1}^n t_k f_i^{(\ell)}(\mathbf{x}_k)\right)^2 \mid \Sigma^{(\ell-1)}, s_+^{(\ell-1)}\right]\right)
$$
$$
= (s_+^{(\ell)})^{1/2} \times \eta_j^{(\ell)},
$$

since $s^{(\ell)}$ and $\tilde{w}_{ij}^{(\ell)}$ are independent and $\eta_j^{(\ell)}$ denotes the normal random variable on the RHS of first line of the above display, which can be seen to be i.i.d. over $j$. Define the matrix $\Sigma^{(\ell)}$ with $\Sigma_{k,h}^{(\ell)} = \mathbb{E}[f_i^{(\ell)}(\mathbf{x}_k) f_i^{(\ell)}(\mathbf{x}_h) \mid \Sigma^{(\ell-1)}, s_+^{(\ell-1)}]$, which is a valid covariance matrix and does not depend on $i$. Thus, since $\Sigma^{(\ell)}$ is a deterministic function of $\Sigma^{(\ell-1)}$ and $s_+^{(\ell-1)}$, we have:

$$
\lim_{M_\ell \to \infty} \phi_{\mathbf{z}_j^{(\ell)} \mid \Sigma^{(\ell)}}(\mathbf{t}) = \exp\left\{-(1/2)(\mathbf{t}^T \Sigma^{(\ell)} \mathbf{t})^{\alpha/2}\right\},
$$

by an application of Lévy's continuity theorem (Billingsley, 1995, Th. 26.3). As such, $\mathbf{z}_j^{(\ell)} \mid \Sigma^{(\ell)}$ is an elliptical $\alpha$-stable random vector.

Next, we proceed with the calculation of $\Sigma^{(\ell)} \mid \Sigma^{(\ell-1)}, s_+^{(\ell-1)}$ in the case of $\tilde{w}_{ij}^{(\ell-1)}$ being standard Gaussian weights and when $g_\delta(\zeta) = \zeta^\delta \mathbf{1}_{\{\zeta > 0\}}$. For this, note that $(z_j^{(\ell-1)}(\mathbf{x}_k), z_j^{(\ell-1)}(\mathbf{x}_h))^T \mid \Sigma^{(\ell-1)}, s_+^{(\ell-1)} \sim \mathcal{N}(0, s_+^{(\ell-1)} \Sigma_*^{(\ell-1)})$, where $\Sigma_*^{(\ell-1)}$ is a two-by-two symmetric matrix with diagonal $(\Sigma_{k,k}^{(\ell-1)}, \Sigma_{h,h}^{(\ell-1)})$, and off-diagonal entries $\Sigma_{h,k}^{(\ell-1)}$. It follows that:

$$
\left(b_j^{(\ell-1)} + z_j^{(\ell-1)}(\mathbf{x}_k), b_j^{(\ell-1)} + z_j^{(\ell-1)}(\mathbf{x}_h)\right)^T \mid \Sigma^{(\ell-1)}, s_+^{(\ell-1)} \overset{i.i.d.}{\sim} \mathcal{N}\left(0, \mathbf{U} + s_+^{(\ell-1)} \Sigma_*^{(\ell-1)}\right),
$$
(5)

where $\mathbf{U}$ is a two-by-two matrix with all entries equal to one, with the independence over the $j$s and conditional on $(\Sigma^{(\ell-1)}, s_+^{(\ell-1)})$. Then for $k, h = 1, \ldots, n$:

$$
\Sigma_{k,h}^{(\ell)} = \mathbb{E}[f_j^{(\ell)}(\mathbf{x}_k) f_j^{(\ell)}(\mathbf{x}_h) \mid \Sigma^{(\ell-1)}, s_+^{(\ell-1)}] = \int_{\mathbb{R}^2} g_\delta(\zeta_1) g_\delta(\zeta_2) p(\zeta_1, \zeta_2) d\zeta_2 d\zeta_1,
$$

where in the second line $p(\zeta_1, \zeta_2)$ is the density of the random vector in Equation (5). It follows that:

$$
\begin{aligned}
\Sigma_{k,h}^{(\ell)} &= \int_{\mathbb{R}} \int_{\mathbb{R}} \zeta_1^\delta \mathbf{1}_{\{\zeta_1 > 0\}} \zeta_2^\delta \mathbf{1}_{\{\zeta_2 > 0\}} p(\zeta_2, \zeta_1) d\zeta_2 d\zeta_1 \\
&= \frac{1}{2\pi v_1 v_2 \sqrt{1-\rho^2}} \int_0^\infty \int_0^\infty \zeta_1^\delta \zeta_2^\delta \exp\left(-\frac{1}{2(1-\rho^2)}\left[\zeta_1^2 v_1^{-2} - 2\rho\frac{\zeta_1 \zeta_2}{v_1 v_2} + \zeta_2^2 v_2^{-2}\right]\right) d\zeta_1 d\zeta_2 \\
&= v_1^\delta v_2^\delta \frac{1}{2\pi\sqrt{1-\rho^2}} \int_0^\infty \int_0^\infty \eta_1^\delta \eta_2^\delta \exp\left(-\frac{1}{2(1-\rho^2)}\left[\eta_1^2 - 2\rho\eta_1\eta_2 + \eta_2^2\right]\right) d\eta_1 d\eta_2,
\end{aligned}
$$

where the second line follows by the bivariate normal p.d.f. where $v_i$ is the standard deviation for the $i$-th entry implied by Equation (5), $\rho$ is the respective correlation and by evaluating the indicator function. In the last line we use the change of variables $\eta_i = \zeta_i / v_i$. This last expression corresponds to a multiple of Equation (15) of Cho & Saul (2009), where $\rho^2 = \cos^2(\theta)$. Explicitly:

$$
v_1^2 = 1 + s_+^{(\ell)} \Sigma_{k,k}^{(\ell-1)}, \ \ v_2^2 = 1 + s_+^{(\ell)} \Sigma_{h,h}^{(\ell-1)}, \ \ \rho = \frac{1 + s_+^{(\ell)} \Sigma_{k,h}^{(\ell-1)}}{v_1 v_2}.
$$

Finally:

$$
\Sigma_{k,h}^{(\ell)} = \pi^{-1} \left[v_1^2 v_2^2\right]^{\delta/2} J_\delta(\theta), \ \theta = \cos^{-1}(\rho),
$$

as required.

## C  Proof of Proposition 2

The expressions for the conditional expectation and variance, $\boldsymbol{\mu}^*$ and $\boldsymbol{\Lambda}^*$ are a direct consequence of the posterior predictive distribution of Gaussian random vectors. We omit the details.

## D  Supplementary algorithms

The following algorithms are used in Algorithm 1. They compute the covariance matrix $\Sigma^{(L)}$ from the random scales. The algorithm used is an independent sample Metropolis–Hastings sampler where we propose from the prior, and accept with probability corresponding to the ratio of the likelihoods.

---

**Algorithm 2** A Metropolis–Hastings sampler for $s_+^{(\ell)} \mid s_+^{(\ell+1)}, \mathbf{y}$

---

**Require:** Scale of the next layers $s_+^{(\ell+1)}, \ldots, s_+^{(L)}$, and $\Sigma^{(\ell-1)}$, the p.d. matrix of the previous layer.

    Propose $s_+^{(\ell),*} \sim S_{\alpha/2}^+$ using the algorithm of Chambers et al. (1976).
    **if** $\ell = 1$ **then**
        Compute $(\Sigma^{(1)})^*$ using the formula from Theorem 1.
    **end if**
    Compute, using the proposed scale, $(\Sigma^{(\ell+1)})^*, \ldots, (\Sigma^{(L)})^*$, with the formulas in Theorem 1.
    Simulate $U \sim Unif(0,1)$.
    **if** $U < p(\mathbf{y} \mid (\Sigma^{(L)})^*)/p(\mathbf{y} \mid \Sigma^{(L)})$ **then**
        Return $s_+^{(\ell),*}$
    **else**
        Return $s_+^{(\ell)}$
    **end if**

---

## E  $\{\Sigma^{(\ell)}\}_{\ell=1}^L$ is consistent under marginalization

Consistency under marginalization means that for $K_1 \sim G(K_0)$, where both $K_1, K_0$ are p.d. $n \times n$ matrices, then a principal $P \times P$ submatrix $K_0^*$ of $K_0$ also induces a distribution $K_1^* \sim G(K_0^*)$, and

---

**Algorithm 3** A Metropolis–Hastings sampler for $s_+^{(L)} \mid \mathbf{y}$

---

**Require:** Vector of observations $\mathbf{y}$, $\alpha$, previous iteration of $s_+^{(L)}$, and $\Sigma^{(L)}$.

    Propose $s_+^{(L),*} \sim S_{\alpha/2}^+$ using the algorithm of Chambers et al. (1976).

    Compute $(\Sigma^{(L)})^* \mid s_+^{(L),*}$, using the formula in Theorem 1.
    Simulate $U \sim Unif(0, 1)$.
    **if** $U < p(\mathbf{y} \mid (\Sigma^{(L)})^*)/p(\mathbf{y} \mid \Sigma^{(L)})$ **then**
        Return $s_+^{(L),*}$.
    **else**
        Return $s_+^{(L)}$.
    **end if**

---

those distributions are consistent, i.e., if we consider the full p.d.f. of $K_1$ and integrate out the entries that do not belong to $K_1^*$ we obtain the same p.d.f. This is a property that is known for Wishart and inverse Wishart matrices (Dawid, 1981).

First, consider the set of positive random scales in the first layer $\mathbf{S}_+^{(1)}$ fixed. Note that $\Sigma^{(1)}$ is a p.d. matrix for any number of observations $n$, since the obtained cross product in Appendix B is a valid covariance function. Similarly, for $P < n$ any $P \times P$ principal submatrix $(\Sigma^{(1)})^*_{1:P,1:P}$ (wlog) is a consistent submatrix, since the random scales are fixed.

Now, consider a principal submatrix (wlog) in the $\ell$ layer: $(\Sigma^{(\ell)})_{1:P,1:P}$, and consider $s_+^{(\ell)}$ fixed. Then $(\Sigma^{(\ell+1)})_{1:P,1:P}$ is a deterministic function of $(\Sigma^{(\ell)})_{1:P,1:P}$, since $s_+^{(\ell)}$ is fixed. This implies that $(\Sigma^{(\ell+1)})_{1:P,1:P}$ is consistent under marginalization, as the only randomness comes from $s_+^{(\ell)}$ over which we can integrate.

## F  PROOF OF PROPOSITION 3

First consider $\alpha < 2$. The posterior of the features is given by:

$$
\begin{aligned}
p(\mathbf{z}_j^{(\ell)} \mid \mathbf{X}, \mathbf{y}) &= \int p(\mathbf{z}_j^{(\ell)}, \{s_+^{(\ell)}\}_{\ell=2}^L, \mathbf{S}_+^{(1)} \mid \mathbf{X}, \mathbf{y}) \prod_{\ell=2}^L ds_+^{(\ell)} \prod_{m=1}^I ds_{m,+}^{(1)} \\
&= \int p(\mathbf{z}_j^{(\ell)} \mid \{s_+^{(\ell)}\}_{\ell=2}^L, \mathbf{S}_+^{(1)}, \mathbf{X}, \mathbf{y}) \\
&\quad \times p(\{s_+^{(\ell)}\}_{\ell=2}^L, \mathbf{S}_+^{(1)} \mid \mathbf{X}, \mathbf{y}) \prod_{\ell=2}^L ds_+^{(\ell)} \prod_{m=1}^I ds_{m,+}^{(1)}.
\end{aligned}
\tag{6}
$$

Employing the elliptical $\alpha$-stable characterization of Theorem 1, the distribution of $\mathbf{z}_j^{(\ell)}$ is given by:

$$
\mathbf{z}_j^{(\ell)} \mid \{s_+^{(\ell)}\}_{\ell=2}^L, \mathbf{S}_+^{(1)} \sim \mathcal{N}(0, s_+^{(\ell)} \Sigma^{(\ell)}).
$$

As such, we have that the posterior of $\mathbf{z}_j^{(\ell)}$ can depend on $\mathbf{y}$ only through the posterior of $\{s^{(\ell)} \mid \mathbf{y}\}$ for $\ell = 2, \dots, L$, since the $\Sigma^{(\ell)}$ is a deterministic function of a fixed design matrix $\mathbf{X}$ and conditional on the random scales $s_+^{(2)}, \dots, s_+^{(\ell)}, \mathbf{S}_+^{(1)}$.

Now, when $\alpha = 2$, all $s_+^{(\ell)}$ are equal to one with probability 1 (i.e., they are degenerate Dirac point mass at one) and therefore the posterior $s_+^{(\ell)} \mid \mathbf{y}$ is also a degenerate point mass at 1. Thus, the posterior of the features cannot depend on the observations when $\alpha = 2$, i.e., in the Gaussian case.

## G  CONDITIONAL MUTUAL INFORMATION

For a bivariate Gaussian vector the mutual information is easily computed by: $-(1/2)\log(1 - \rho^2)$, where $\rho$ is the correlation coefficient between the two entries of the vector (Cover & Thomas, 2006,

p. 252). However, recall that in our non-Gaussian setting, marginal moments or correlations are not defined and we need a conditional construction. Consider three random variables $Y_1, Y_2, S$, the conditional mutual information (Cover & Thomas, 2006, p. 23) is given by

$$I(Y_1; Y_2 \mid S) = \int_{\mathcal{S}} \int_{\mathcal{Y}_1 \times \mathcal{Y}_2} \log \left( \frac{p(y_1, y_2 \mid s)}{p(y_1 \mid s)p(y_2 \mid s)} \right) p(y_1, y_2 \mid s)p(s)dy_1 dy_2 ds$$

$$= \int_{\mathcal{S}} D_{KL}[p(Y_1, Y_2 \mid s) \mid\mid p(Y_1 \mid s)p(Y_2 \mid s)]p(s)ds,$$

where $D_{KL}$ is the Kullback–Leibler divergence. In our case, we consider $Y_1, Y_2$ as two observations from the deep $\alpha$-kernel process, and $S$ corresponds to the conditional scales. Since given $S$ the observations are conditionally-Gaussian, we can apply the formula that employs the correlation and integrate over the scales to obtain the mutual information. Specifically, we use:

$$I(Y_1; Y_2 \mid S) = -(1/2) \int_{\mathcal{S}} \log(1 - \rho_{Y_1, Y_2}^2(s))p(s)ds.$$

To perform the integration we use Monte Carlo samples of $\{s_+^{(\ell)}\}_{\ell=2}^L, \mathbf{S}_+^{(1)}$, and use the correlation induced by the $\Sigma^{(L)}$.

## H  ADDITIONAL NUMERICAL RESULTS FOR SIMULATED DATA

### H.1  TIMING OF SIMULATIONS

In Table 5 we display the running time for the six methods employed in the simulations described in Section 3. All the times are in seconds and display CPU times. We avoid any type of parallelization throughout our experiments.

Table 5: Average (SD) time (s) for the six methods in the numerical examples for one, two, and ten dimensions.

| Method | 1D | 2D | 10D |
|---|---|---|---|
| D$\alpha$-KP | 181.23 (15.3) | 200.19 (31.16) | 9713.51 (212.10) |
| DIWP | 206.56 (10.24) | 312.29 (21.17) | 14509.31 (2783.10) |
| GP Bayes | 17.26 (0.44) | 21.35 (0.24) | 729.78 (26.96) |
| GP MLE | 0.08 (0.00) | 0.18 (0.01) | 49.89 (2.51) |
| NNGP | 131.03 (11.99) | 167.85 (6.36) | 11044.99 (1927.03) |
| Stable | 33.14 (5.64) | 1456.72 (85.45) | – |

### H.2  COVERAGE OF CREDIBLE INTERVALS

In Table 6 we display the percent of observations in the out-of-sample observations captured by the 90% credible intervals produced by the Bayesian methods. The GP MLE method is a frequentist method and as such does not produce valid credible intervals. The Stable method is not available for 10 dimensions. The methods that use stable random variables are the only ones that consistently manage to be close to the nominal coverage of 90%. The other methods typically have coverage much below 90%, especially in high dimensions.

### H.3  VISUALIZATIONS OF SIMULATION RESULTS

### H.3.1  PLOTS FOR TWO DIMENSIONS

Figure 4 displays the function fit for the example in two dimensions and the residuals with the accompanying credible intervals. We do this for the first split of the dataset. Recall that the true function is $f(\xi_1, \xi_2) = 5 \times \mathbf{1}_{\{\xi_1 > 0\}} + 5 \times \mathbf{1}_{\{\xi_2 > 0\}}$. The D$\alpha$-KP and the Stable method are the only ones that are able to capture the jump without oversmoothing, and with narrower credible intervals.

Table 6: Average (SD) percent of out-of-sample observations captured by the 90% credible intervals in the twenty splits of the simulation examples.

| Method | 1D | 2D | 10D |
|---|---|---|---|
| D$\alpha$-KP | 90.85% (5.12) | 92.47% (5.63) | 89.70% (2.42) |
| DIWP | 92.70% (1.89) | 95.86% (1.22) | 60.47% (3.08) |
| GP Bayes | 94.85% (2.43) | 94.14% (3.02) | 65.17% (5.82) |
| GP MLE | – | – | – |
| NNGP | 92.50% (1.79) | 95.99% (1.12) | 60.65% (2.82) |
| Stable | 91.65% (4.40) | 91.11% (4.63) | – |

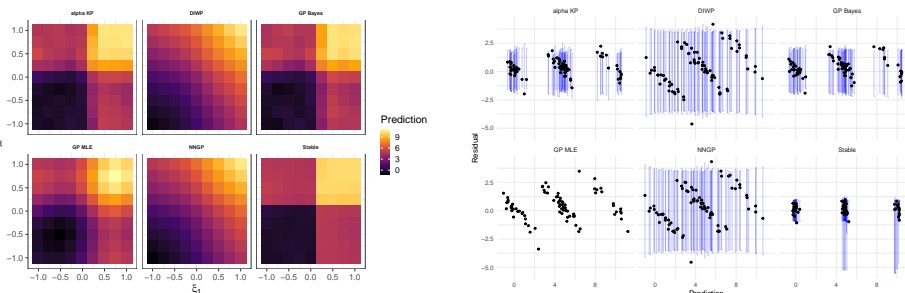

(a) Function fit for the different methods.    (b) Residuals versus predictions for the methods that produce valid credible intervals.

Figure 4: Function fit and residuals with 90% credible intervals for the six methods in the simulations in two-dimensions.

### H.3.2 PREDICTIONS AND UNCERTAINTY QUANTIFICATION IN HIGH DIMENSIONAL SIMULATION

In Figure 5 we display the a scatter plot of residuals (observation - prediction) with the prediction on the first split of the simulation example for 10 dimensions. Note that the D$\alpha$-KP and GP MLE have the smallest range of residuals, while the other methods have a wider range of residuals.

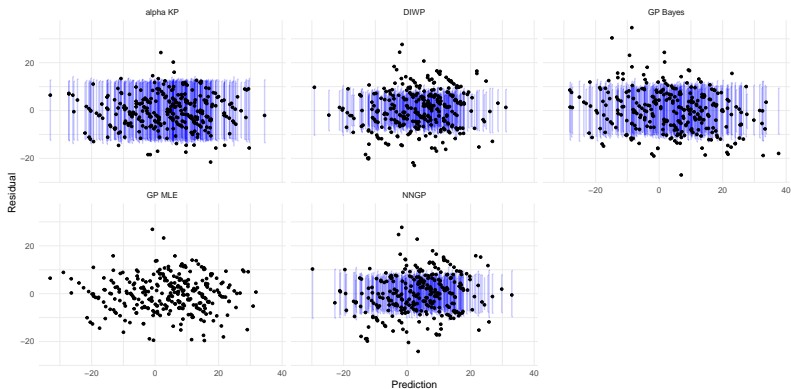

Figure 5: Residual plot for first split of the high dimensional example with 90% credible intervals.

### H.3.3 ABLATION STUDY ON $\alpha$ FOR THE ONE-DIMENSIONAL EXAMPLE

In Figure 6, we display the predicted values (solid lines), along with the 25th to 75th percentiles for the posterior predictive intervals (shaded regions) on a finer grid (1000 points) for the example in one dimension considered in Section 3.2, for $\alpha = 0.5, 1, 1.5, 1.95, 1.99$. It is apparent that values of

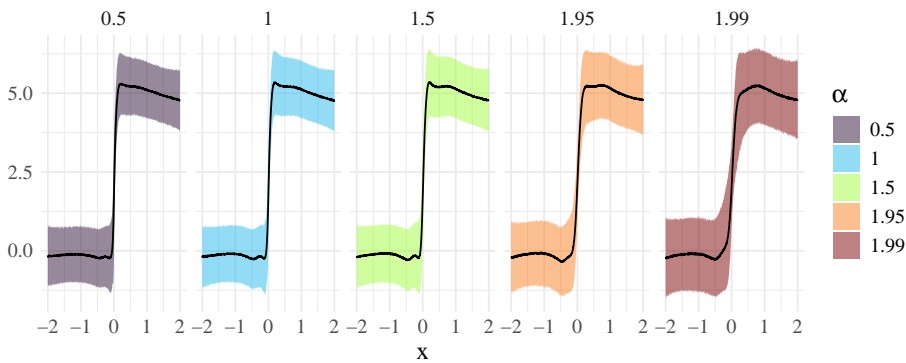

Figure 6: Comparison of predictions (solid lines) and 25th to 75th percentile posterior predictive intervals (shaded regions) in one dimension for different values of $\alpha$ for the D$\alpha$KP.

$\alpha$ closer to 2 (the Gaussian limit) tend to oversmooth the fit around the jump discontinuity at zero and in general give wider posterior predictive intervals.

### H.3.4 POSTERIOR DISTRIBUTION OF THE KERNEL IN THE LAST LAYER FOR EXAMPLES IN ONE AND TWO DIMENSIONS

In Figure 7 we display the quantiles of the posterior of the kernel matrix in the last layer for the one and two dimensional examples shown in the main text. Recall that the training points are on equally-spaced grids. Figure 7 serves as evidence that the kernel is indeed stochastic, with a non-degenerate posterior. Furthermore, the values obtained in the 75th percentile are quite high relative to the median and is an indication of a heavy-tailed distribution.

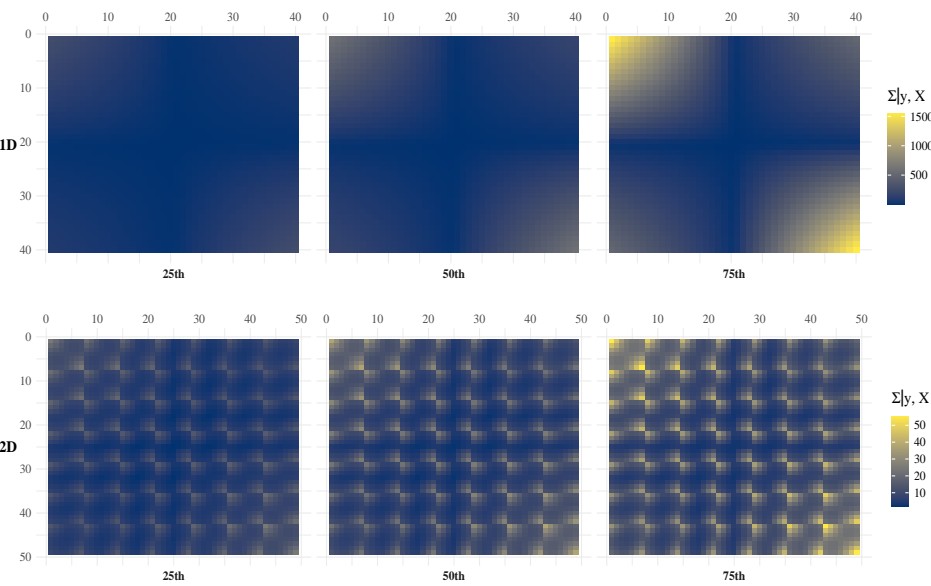

Figure 7: Posterior quantiles (*left*: 25, *center*: 50, and *right*: 75) of the distribution of the kernel in the last layer for the examples in 1 dimension (*top*), and 2 dimensions (*bottom*).

# I ADDITIONAL NUMERICAL RESULTS FOR UCI DATA

Figures 8, 9 and 10 displays a scatter plot of the predictions and residuals (observation - prediction) of the test set in the first split of the UCI datasets, as well as the credible intervals for said observations. The predictions of the D$\alpha$-KP are much closer to the observations, while the DIWP and NNGP present worse predictions relative to the observations. Moreover, the width of the credible intervals of the D$\alpha$-KP vary, while the DIWP and NNGP are much more uniform. The results indicate that the credible intervals of the D$\alpha$-KP capture more observations than the competing methods. Figure 11 displays the trace plot for the log-likelihood of the fit of the deep $\alpha$-kernel process in the first split of the Boston dataset, indicating a fast mixing.

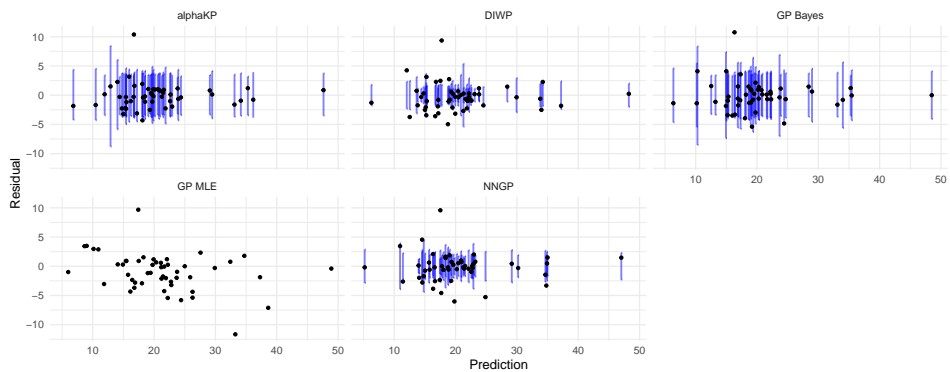

Figure 8: Comparison of predictions and observations in the test set for the first split of the Boston dataset, with $90\%$ credible interval.

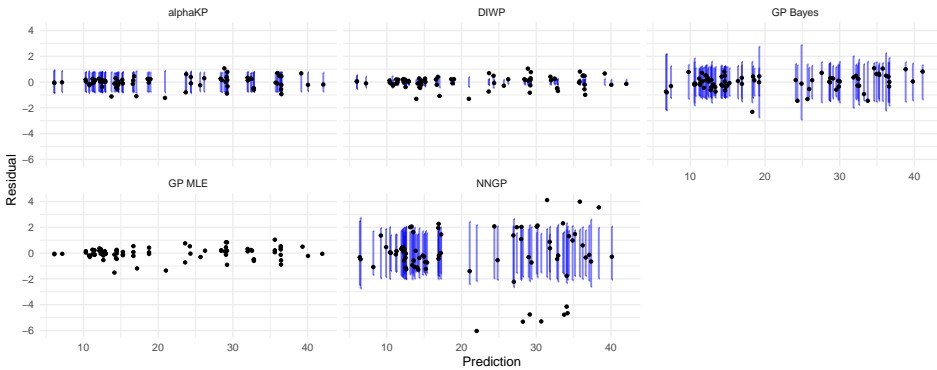

Figure 9: Comparison of predictions and observations in the test set for the first split of the Energy dataset, with $90\%$ credible interval.

## I.1 TIMING OF UCI EXPERIMENTS

In Table 7 we display the average running time for each of the methods, except the Stable method as the input dimensions of the data sets are all greater than 2. Overall the GP MLE and GP Bayes are faster than the kernel process methods. The kernel process methods have a similar running time in each dataset.

## I.2 COVERAGE OF CREDIBLE INTERVALS FOR UCI DATASETS

In Table 8 we display the average coverage of the 90% credible intervals for the UCI datasets in the same splits used in Section 4. The D$\alpha$-KP consistently is close to the nominal $90\%$ coverage, while the other methods do not have a consistent coverage close to the nominal 90% percent.

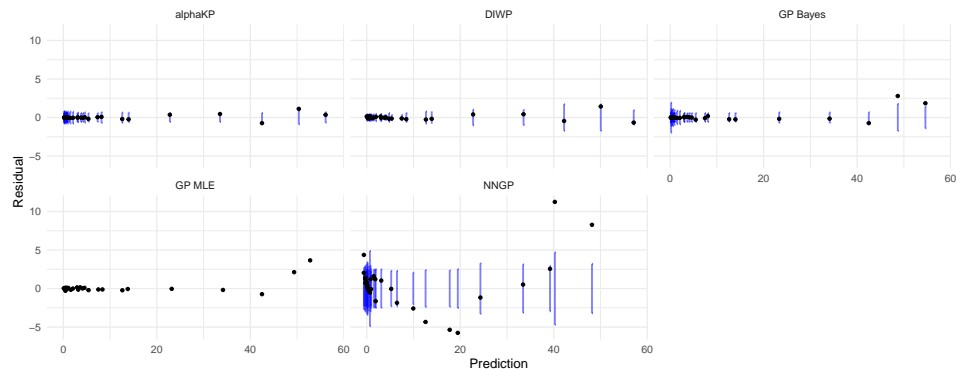

Figure 10: Comparison of predictions and observations in the test set for the first split of the Yacht dataset, with 90% credible interval.

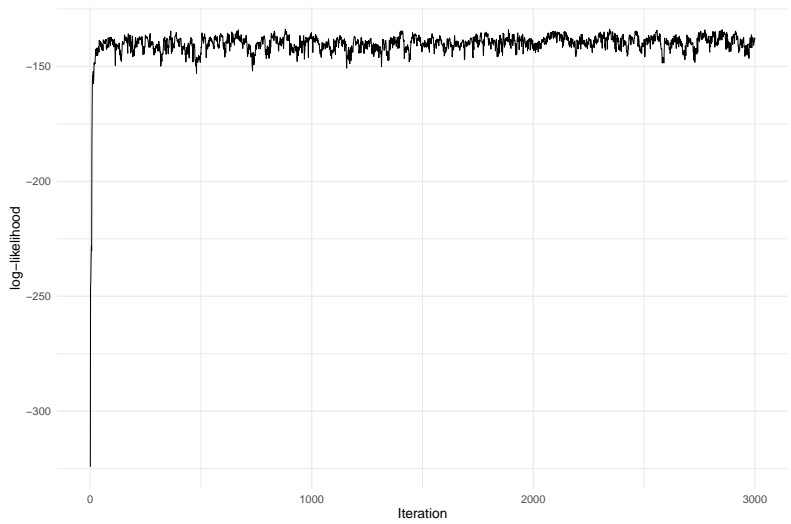

Figure 11: Traceplot of the log-likelihood for the first split of the Boston dataset.

Table 7: Average (SD) time (s) for the methods in the three UCI datasets over twenty different splits. Times for the Stable method are not available.

| Method | Boston | Energy | Yacht |
|---|---|---|---|
| D$\alpha$-KP | 22320.89 (690.23) | 36154.63 (1812.99) | 4590.63 (149.31) |
| DIWP | 20104.97 (3877.92) | 41291.76 (15973.11) | 22177.2 (10017.53) |
| GP Bayes | 1047.13 (6.51) | 2151.64 (12.37) | 271.82 (2.21) |
| GP MLE | 142.06 (6.68) | 390.52 (9.36) | 22.18 (1.20) |
| NNGP | 18242.54 (8474.21) | 36872.04 (21713.89) | 9177.82 (3803.24) |
| Stable | – | – | – |

## J    FEATURE LEARNING IN ADDITIONAL DATASETS

In Figures 12 and 13 we provide the scatter plot of the features of the Boston and Yacht data sets under D$\alpha$-KP, along with the corresponding box plots and q-q plots for each of the features. Similar to the Energy dataset, the Boston features display a heavy-tailed behavior, while those for Yacht are closer to normality.

Table 8: Coverage (SD) of the 90% credible intervals for the methods in the UCI datasets over twenty different splits. Stable method is not available, and GP MLE does not produce credible intervals.

| Method | Boston | Energy | Yacht |
|--------|--------|--------|-------|
| D$\alpha$-KP | 91.86% (5.14) | 87.40% (3.91) | 95.32% (2.86) |
| DIWP | 57.35% (7.50) | 48.77% (6.05) | 90.00% (6.61) |
| GP Bayes | 93.24% (4.43) | 97.01% (1.79) | 95.00% (3.22) |
| GP MLE | – | – | – |
| NNGP | 68.53% (8.26) | 68.77% (5.45) | 67.58% (7.65) |
| Stable | – | – | – |

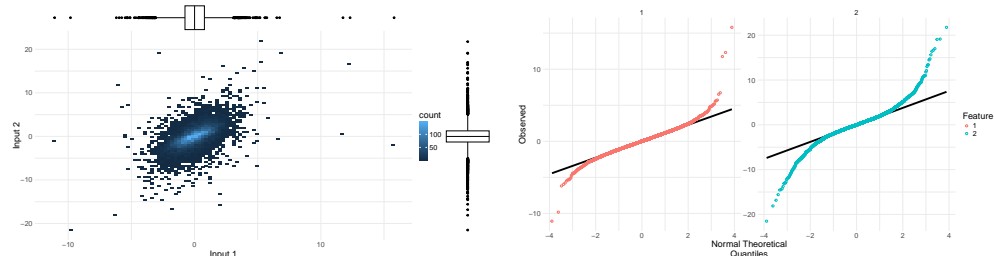

(a) Scatterplot of the first two features, with box plots for the marginals.

(b) Normal q–q plot of the first two features.

Figure 12: Posterior distribution of the features for the Boston data set using 10k posterior MCMC samples with 1k burn-in samples.

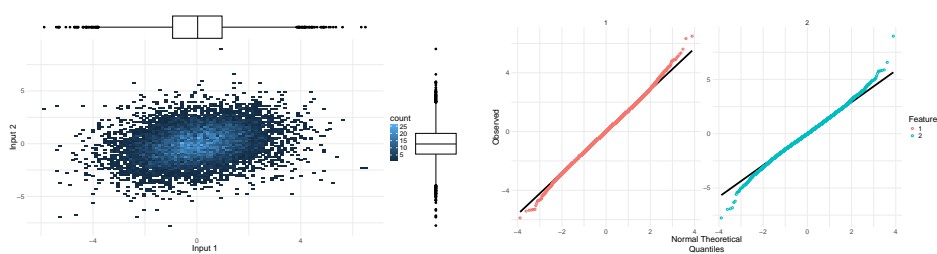

(a) Scatterplot of the first two features, with box plots for the marginals.

(b) Normal q–q plot of the first two features.

Figure 13: Posterior distribution of the features for the Yacht data set using 10k posterior MCMC samples with 1k burn-in samples.

