# OpenReview forum: "Deep Kernel Posterior Learning under Infinite Variance Prior Weights"
_ICLR.cc/2025/Conference — ICLR 2025 Poster_

### Official Review · Reviewer_o61u · 2024-10-29

**Soundness:** 3
**Presentation:** 3
**Contribution:** 3
**Rating:** 8
**Confidence:** 2

**Summary:**

This paper considers an infinite-width Bayesian neural network, with infinite-variance weights.  Interestingly, in this setting, representations/kernels do not become deterministic as usual in infinite-width limits, but remain stochastic, and hence learnable.

**Strengths:**

The theoretical contributions are extremely strong.  They consider an extremely interesting regime, and the question of how to avoid deterministic kernels in infinite-width networks is a critical one for Bayesian neural networks.

They have an extensive and excellent Appendix.

**Weaknesses:**

The experiments are weaker, but this can be expected in a paper with a mainly theoretical focus and contribution, such as this one.

**Questions:**

N/A

---

> ### Author Response · Authors · 2024-11-16
> **Response to  o61u**
>
> We are delighted by your enthusiastic review, and thankful that you found our paper (in your own words) *extremely strong*, *extremely interesting* and that even the Appendix has received your praise!
>
> In the light of your own readings, as well as our responses to the other reviewers and their reception of the paper, we respectfully urge you to reconsider your score to raise it above a "marginal accept" level of 6. We also urge you to reconsider if our responses and clarifications have been successful in boosting your confidence in your assessment of our paper.

---

> > ### Comment · Reviewer_o61u · 2024-11-26
> > **Rs**
> >
> > Thanks for the reviewers comments.  I have raised my score.

---

> > > ### Author Response · Authors · 2024-11-26
> > >
> > > We sincerely thank you for your positive reassessment and for your willingness to raise your score!

---

### Official Review · Reviewer_xtbp · 2024-11-01

**Soundness:** 3
**Presentation:** 2
**Contribution:** 3
**Rating:** 8
**Confidence:** 3

**Summary:**

Although deep kernel processes allow for more flexibility than standard GPs, they still assume smoothness and finite variance, which is unsuitable for modelling data with heavy tails or discontinuities. This is particularly relevant to representation learning, where deterministic covariance kernels limit the ability to learn non-degenerate, data-dependent representations. In contrast, $\alpha$-stable distributions are well-suited to problems were large fluctuations or discontinuities can be expected. Taking inspiration from these strengths, in this paper the authors assign $\alpha$-stable distributions with infinite variance to the weights in each layer of a BNN, such that the BNN now converges to a deep $\alpha$-stable kernel process rather than GP in the infinite-width limit.

They also develop conditional Gaussian representations of $\alpha$-stable processes that allow for recursive computation of the stochastic kernels at each layer. Through a combination of numerical experiments on synthetic discontinuous functions and evaluations on widely-used datasets from the UCI repository, the authors demonstrate that the deep $\alpha$-stable kernel process outperforms competing approaches when modelling discontinuous or heavy-tailed data. However, simpler models or GPs with traditional Gaussian priors might be preferable for smoother data due to their lower computational cost.

**Strengths:**

- Due to my limited familiarity with α-stable processes, I didn’t review all of the theoretical details. However, the paper reads as well-structured and complete, and the writing clearly highlights the contributions in the context of prior work with a solid mix of theoretical grounding, ablation studies, and performance benchmarks on public datasets.
- The problem statement is otherwise well-described, and I think this work should be of interest to future work on BNNs by addressing core limitations of smoothness and finite variance in existing models. I also wonder whether the non-Gaussian priors explored in this work could also be considered in other deep models?
- I appreciated the extensive numerical experiments that showcased the model’s performance in controlled settings that properly validate it meets the intended criteria of modelling non-Gaussian behaviours. I do wish the section real-world datasets was more compelling however.

**Post-rebuttal update**

Raised score from 6 to 8.

**Weaknesses:**

Although the paper is well-written overall, the dense presentation of different concepts makes it challenging to intuitively grasp without visual aids or simplified examples. I would recommend the inclusion of more figures illustrating how the α-stable kernel process differs from traditional GPs, ideally showing its behavior on simple synthetic data to highlight the flexibility and stochasticity introduced by their method. Some detail can likely also be moved to the appendix instead.

**Questions:**

N/A

---

> ### Author Response · Authors · 2024-11-16
> **Response to xtbp**
>
> **Strengths:**
>
> - *I also wonder whether the non-Gaussian priors explored in this work could also be considered in other deep models?*
>
> We believe the answer is yes! Please see our responses to Reviewer Qsid, who suggested a non-trivial possibility of adapting the current work to the neural tangent kernel (NTK) regime. We have summarized our detailed response in the "Overall summary of the changes to the manuscript," point number 3, and in the Concluding Remarks (Section 5) of the revised manuscript.
>
> **Weaknesses:**
>
> We thank you for your overall positive comments on the paper, especially on writing. We accept your suggestion of adding more visual aids to help understanding. The revised manuscript contains two additional figures. **Figure 6** shows the model fit and uncertainty quantification in the 1-d example as a function of $\alpha$ (following also a request from Reviewer a3tm). This shows that the model can capture jump type discontinuities better at low $\alpha$, and gives smoother fit under the Gaussian limit when $\alpha\to 2$. We have also added **Figure 7**, that shows the posterior of the *random kernel* learned from the data at the 25th, 50th and 75th percentiles, which is a key difference from the *degenerate deterministic posterior* kernel in the GP limit, which cannot be learned from the data.
>
> We note further that our real data examples, in terms of sample size $n$ and input dimension $I$ are at par with recent competing literature on GP models, and that they serve to demonstrate the main contributions of the paper. Hence, we have chosen to leave the real data section unchanged.
>
> In the light of our responses to your and other reviewers' questions, we respectfully urge you to reconsider your score, i.e., whether you still truly feel that this is a ``marginal accept'' paper.

---

> > ### Comment · Reviewer_xtbp · 2024-11-26
> > **Acknowledgement of rebuttal**
> >
> > Thank you for your detailed responses across all reviews. In light of the rebuttals, and after skimming the updated version of the paper, I am also lifting my score up to 8.

---

> > > ### Author Response · Authors · 2024-11-26
> > >
> > > We sincerely thank you for your positive reassessment and for your willingness to raise your score!

---

### Official Review · Reviewer_3kZC · 2024-11-03

**Soundness:** 2
**Presentation:** 2
**Contribution:** 2
**Rating:** 5
**Confidence:** 2

**Summary:**

The article presents a deep kernel process with infinite variance priors including a recursive formula and experiments illustrating dependence on the inputs and behavior for discontinuous targets.

**Strengths:**

The article
* provides a discussion of references, theoretical results, and experiments.
* discusses potential benefits in prediction and uncertainty quantification.
* suggests feature learning that is not possible under a Gaussian process.

**Weaknesses:**

1. My main concern with the article is the writing and presentation, which I think need to be improved.
The abstract gives a long discussion of prior works but ideally it should instead give a crisp description of the main points in the article. The lengthy discussion in the introduction comments on prior works and perceived limitations, but does not provide a sufficiently concise and clear description of the objective, motivation, and contributions of the present work. Terminology could be introduced more clearly in preliminaries, rather than in prior works. The writing is in parts repetitive and the thread is not sufficiently clear.  \
&nbsp;
\
Suggestions proposed by the ICLR Associate Program Chair:
    1. Revise the abstract to focus on succinctly stating the key contributions and results of the paper, rather than discussing prior work.
    2. Restructure the introduction to clearly state the objectives, motivation and main contributions upfront before discussing related work.
    3. Add a preliminaries section to introduce key terminology and concepts.
    4. Review the paper to remove repetition and improve the logical flow of ideas between sections.  \
&nbsp;

2. My second concern is the motivation and innovations, which I do not find sufficiently clearly articulated in the article.
For Theorem 1 it is mentioned that the result permits characterization of the deep kernel process in an infinite variance setting and for Proposition 3 that it results in a posterior distribution of the features that depends on the observations. I can see that there are differences to prior works and that feature learning is an interesting topic, but in my opinion it is not sufficiently clearly explained what the motivation is for formulating this particular deep kernel process. Another point that is not described in sufficient clarity is what the concrete and significant conceptual and technical innovations are that go into obtaining the proofs and how they could be useful for obtaining further relevant advances.  \
&nbsp;
\
Suggestions proposed by the ICLR Associate Program Chair:
    1. Provide a clearer explanation of why an infinite variance setting is important or beneficial for deep kernel processes.
    2. Discuss more explicitly how the ability to learn features from observations addresses limitations of previous approaches.
    3. Highlight the key technical challenges in proving Theorem 1 and Proposition 3, and how overcoming these challenges advances the field.
    4. Explain potential applications or extensions of their technical innovations beyond this specific model.
    5. Provide a clearer explanation of why this particular deep kernel process formulation was chosen and what advantages it offers.  \
&nbsp;

3. Description of the experiments.
In my opinion the motivation and description of the experiments are not sufficiently clearly presented.  \
&nbsp;
\
Suggestions proposed by the ICLR Associate Program Chair:
    1. Provide clearer motivation for each experiment, explaining how it relates to the theoretical results.
    2. Improve the description of experimental setups, including details on datasets, baselines, and evaluation metrics.
    3. Discuss the implications of the experimental results in more depth, relating them back to the theoretical contributions.

**Questions:**

* Could the authors please offer more comments on the motivation for considering this particular model?
* Could the authors please offer comments on what the concrete and significant technical advances are?

---

> ### Author Response · Authors · 2024-11-16
> **Response to 3kZC**
>
> **Weaknesses:**
>
> We are afraid that we are unable to locate in your long list of weaknesses what, precisely, the issues are. It is also not quite clear to us exactly what changes in writing you are asking for. Other reviewers appear more or less satisfied with the writing and presentation, and although some have found the paper technical (which might have to do with its subject matter), we do not believe they have questioned its clarity outright. Hence, after thorough considerations, have chosen to retain the main structure of the paper. We also hope our responses to the other reviewers are helpful in your re-assessment, and if you have any specific questions (beyond a generic "not sufficiently clear" and such), we are happy to consider those during the rebuttal period. We also considered the suggestions listed under "ICLR Associate Program Chair" but despite our efforts, we are unable to locate anything concrete and actionable.
>
> **Questions:**
>
> - **Motivations:** We believe this is well articulated in the paper, but for your convenience, we repeat the main motivations here. The motivation is that this model arises as the infinite-width limit of deep Bayesian neural networks under quite general infinite variance prior weights. Thus, our results provide a convenient approach for posterior uncertainty quantification, which was so far was only possible for neural network GP (NNGP) models. Furthermore, there are nontrivial differences with the NNGP limits for our result: (1) We are able to perform representation learning that a NNGP (with a deterministic kernel) is unable to learn. (2) Under discontinuous true functions, we are able to obtain much better statistical performance (lower MSE and narrower credible intervals) compared to GP models, which tend to oversmooth.
>
> - **Significant Advances:** The main advances and contributions are  succinctly summarized in Section 1.2. The key technical advance is the deep stable process itself, as developed in the main result of our paper in Theorem 1, which allows representation learning through a random (conditional) covariance kernel in each layer.

---

> > ### Comment · Reviewer_3kZC · 2024-11-26
> > **Re: Response**
> >
> > The author responses did not take concrete steps to address the concerns raised in the original review. Taking a look at other reviewers' comments, I am inclined to think that the results are of interest. I am torn between maintaining the score or not. I am lowering my confidence and will follow up with other reviewers during the reviewer-AC period. At any rate, I encourage the authors to consider taking steps to make the text more accessible and better communicate the main message, technical significance, and relevance, as suggested in the initial review.

---

### Official Review · Reviewer_Qsid · 2024-11-04

**Soundness:** 3
**Presentation:** 3
**Contribution:** 3
**Rating:** 8
**Confidence:** 4

**Summary:**

**Summary**
Neal's infinite width neural networks, even when extended to the deep setting, do not account for representation learning. That is, they result in a GP with a fixed covariance function. This is unlike typical but intractable Bayesian neural networks, which are capable of representation learning. The authors consider heavy tailed priors, elliptically distributed with infinite variance, and show that the network converges to a process with $\alpha$-stable marginals. Each layer is conditionally Gaussian, even though the covariances are not necessarily even defined.  The main technical tool is the CLT for alpha-stable RVs (Theorem 4 from another paper, with alpha=2 being the classical CLT), and a (conditionally Gaussian) Gaussian mixture representation of the characteristic function of the alpha-stable RV (equation (1)). Furthermore, a computationally viable MCMC method for doing inference is also provided. The derived method is demonstrated on some UCI benchmarks, showing better prediction than other methods, and being faster than a previous heavy-tailed method.

**Strengths:**

**Strengths:**
- The proposed model is capable of representation learning. Specifically, Proposition 3 nicely shows that the feature at layer l depends on the data X, y observed in training, which is unlike many finite width models. It is nice to have this notion of representation learning formalised in this simple way.
- The idea of the paper is relatively straightforward: everything is conditionally Gaussian given this scale parameter, which induces the heavy-tailed behaviour. I see this simplicity as a strength, as it doesn't require complicated constructs to convey a nice and useful result.
- The method is demonstrated, firstly on some artifical but enlightening discontinuous functions (where it either almost matches the shallow stable process or is able to compute a posterior where the shallow stable process fails), and also on some UCI benchmarks, where it usually outperforms other methods and the previous shallow stable process is intractable. There are also some nice graphics comparing the posterior predictives of other models in 1D and a QQ plot showing non-Gaussian behaviour of the features. More extreme non-Gaussian behaviour is shown in the appendix (in particular, Figure 10 - I am wondering why the authors did not choose to include this as their main non-Gaussian demonstration in the main paper?)
- Appendix B and D, which contain the proofs, appear to not have any major technical errors.

**Weaknesses:**

**Weaknesses:**
- Unless I am mistaken, "The key finding is that the conditional mutual information decays at a slower rate for smaller α" should be "The key finding is that the mutual information [which itself is computed via MCMC as an expected conditional mutual information] decays at a slower rate for smaller α." Figure 1 shows a mutual information, not conditional mutual information.


**Minor:**
- Theorem 1. $J_\delta(\theta)$ can be computed explicitly, and this is obvious to people familiar with the result of Cho & Saul (as mentioned in the following paragraph). But some readers might find it alarming to see $J_\delta(\theta)$ before it is defined. Perhaps the result of Cho & Saul could be made more explicit before theorem 1 (i.e. move $J$ up in the page)
- Figure 1 and 2 labels are too small. Sans-serif font clashes with the text in the main paper.
- last sentence of first paragraph of section 4.2 seems to have odd grammar.

**Questions:**

**Questions:**
- I don't understand this sentence, could you please clarify? "The third limitation is one noted by Neal (1996), that in the case of multiple outputs, GPs cannot learn the covariance structure, as the covariance is always zero under a Gaussian limit." Are you saying that for a vector-valued neural network output, any two outputs are independent? This would be false, even in the finite variance setting, as long as one considers non-isotropic Gaussian weights (which is relatively straight-forward). E.g. see weights W in section 1 of "Avoiding Kernel Fixed Points: Computing with ELU and GELU infinite networks, AAAI 2021".
- Another related work might be "Richer priors for infinitely wide multi-layer perceptrons". In particular, Theorem 1 and Proposition 2 seem to show that the resulting posterior predictive distribution also marginalises out an $\alpha$-stable distributed scale matrix in the covariance matrix. This seems similar to equation (19), where a scale and a location parameter are integrated out in the posterior predictive, and the resulting model is a kind of infinite mixture of GPs. Although I think the motivation for considering such infinite mixtures is different: this paper focusses on heavy tailed whereas the other one focusses on exchangeability, and considers more "mild" mixing. Still, it could be nice to motivate the current approach both from the perspective of heavy-tailed priors and from the perspective of exchangeability in the hidden layers.
- Is it possible to construct deep $\alpha$-stable processes for gradient flow (extending NTK), rather than BNNs?
- As far as I can tell from the proof of Theorem 1 in Appendix B, rectified monomial activations are not strictly required for the proof to work. All that is required is that a recursive equation for the conditional covariance is available, which is true beyond this activation for other activations (see e.g. the works I mentioned above). It might be nice to extend the theoretical statement to more broad activation functions.

---

> ### Author Response · Authors · 2024-11-16
> **Response to Qsid**
>
> We thank Reviewer Qsid for a favorable review, nuanced questions, and for thought-provoking suggestions that open up new possibilities for building upon the current work, such as adapting it to the NTK regime.
>
> **Strengths:**
>
> *in particular, Figure 10 - I am wondering why the authors did not choose to include this as their main non-Gaussian demonstration in the main paper?*
>
> **Response:** There was no particular reason for this, as we analyzed 3 data sets and reported results for all three of them (one in main and two in supplement). But we agree with you Figure 10 makes the non-Gaussianity in features even clearer. We have swapped the previous Figures 3 and 10 in the revised manuscript.
>
> **Weaknesses:**
>
>  -   You are not really mistaken, but the issue here is with the (rather unfortunate) definition of conditional mutual information (CMI) itself, which Cover and Thomas (2006, p. 23) define as:
> \begin{align*}
>  I(Y_1; Y_2 \mid S)
>  &= \int_{\mathcal{S}} D_{KL}[p(Y_1,Y_2\mid s) || p(Y_1\mid s)p(Y_2\mid s)] p(s)ds.
> \end{align*}
> We reproduced this definition in Appendix G. As you can see, the RHS has $S$ integrated out, but the LHS uses the notation $I(Y_1; Y_2 \mid S)$. Thus, what Cover and Thomas mean by conditioning is actually a mixture with respect to $S$ (which is why we call this notation unfortunate), and what we report in Fig. 1 is actually CMI (computed by Monte Carlo integration), not MI (according to the Cover and Thomas terminology). Nevertheless, the MI in the caption of Fig. 1 should have been CMI, and it is now corrected.
>
> **Minor:**
>
> -  We understand your point, but it is rather hard to move $J_\delta(\theta)$ up before Theorem 1, because $\theta$ is yet to be defined. We have modified the statement of Theorem 1 to make a reference to Eq. (4) of Cho and Saul (2009), where $J_\delta(\theta)$ is precisely defined.
>
> -  In the revised pdf, we have increased the size of the font labels and have used Serif (Times) fonts for Figs. 1 and 2.
>
> -  There was a period that should have been a comma. The last two sentences have now been merged, and the revised sentence now reads: "*The non-Gaussianity of $\mathbf{z}_j^{(\ell)}$ also serves as confirmation that $\Sigma^{(\ell)}$ is indeed stochastic, since otherwise we would obtain Gaussian features.*"
>
> **Questions:**
>
> -  We apologize for the confusion. We paraphrased the following quotation from "Chapter 2. Priors for Infinite Networks," by Neal (1996, p. 34) that reads: "*It is easy to see, however, that the covariance between $f_{k_1} (x^{(p)})$ and $f_ {k_2}(x^{(q)})$ is zero whenever $k_1\ne k_2$ since the weights into different output units are independent under the prior.*" The point here is that under the model of Eq. (2)--(3) of our paper, and in case of multiple outputs, if all $w_{kj}^{(L)}$ terms are i.i.d. Gaussian then the limits are also independent, and hence Neal's claim. But  now suppose  $w_{kj}^{(L)}$ and $w_{k'j}^{(L)}$ both depend on the same $s_+^{(L)}$. Then marginally, they are merely exchangeable and not independent. Of course, you can capture dependence across $k$ even in a GP using a valid cross-covariance model (as you have suggested above), but it won't appear under i.i.d. Gaussian weights, whereas it comes for free in our setting of *conditionally Gaussian exchangeable weights.* We have rewritten the sentence as: ``*The third limitation is one noted by Neal (1996), that in the case of multiple outputs, GPs cannot learn the covariance structure under i.i.d. weights, as the covariance is always zero under an isotropic Gaussian limit.*"
>
> - Thank you for pointing out this paper to us and we agree that Eq. (19) bears similarities with what we are trying to do. As pointed out in response to your last question, exchangeability *comes for free* in our Gaussian scale mixture regime. The idea of the paper you refer is similar, although the motivation is different. We have included a reference to it in the revised manuscript, Section 2, before Theorem 1.
>
> -  We believe your intuition is correct! Several works have found empirical evidence that SGD noise could be non-Gaussian and heavy-tailed, see, e.g., ``A tail-index analysis of stochastic gradient noise in deep neural networks'' by Simsekli et al. (2019, ICML). We believe a construction similar to ours could be used to study the non-Gaussian NTK limit as well, and the chief difference would be mixtures of NTK kernels instead of mixtures of Cho and Saul kernels. We now mention this possibility under Section 5 as a potential future work.
>
> -  Yes, you are also correct on this. We used the rectified monomial activation because it is sufficient for our purposes, and the resulting Cho and Saul formulas are simple. But the main construction of the deep kernel process does not depend on it. We now mention using other activations as a potential future work in Section 5, and have added a reference to ``Avoiding Kernel Fixed Points: Computing with ELU and GELU infinite networks, AAAI 2021.''

---

> > ### Comment · Reviewer_Qsid · 2024-11-17
> >
> > Thanks for responding to my questions and suggestions. The revisions look good to me.

---

> > > ### Author Response · Authors · 2024-11-22
> > >
> > > We thank Reviewer Qsid for favorable comments on the revisions. We also wanted to check with all other reviewers if they had any pending questions or concerns not addressed in our rebuttals.

---

### Official Review · Reviewer_a3tm · 2024-11-06

**Soundness:** 3
**Presentation:** 2
**Contribution:** 3
**Rating:** 5
**Confidence:** 2

**Summary:**

The authors consider Bayesian neural networks in the infinite-width limit under infinite variance priors on the weights. They show that the resulting covariance kernel becomes a random process that depends on the given data, in contrast to the finite variance prior case, where the kernel is deterministic. Additionally, they demonstrate that the representation in each layer is Gaussian when conditioned on the associated covariance realization. In the experiments, they show that the data-dependent stochastic covariance enables feature learning, and that the resulting predictive Gaussian process achieves improved performance on benchmarks.

**Strengths:**

- The technical aspects of the paper appear solid, although I have not checked the results in detail.
- The experiments seem to effectively support the theoretical claims.

**Weaknesses:**

- The paper is somewhat challenging to approach, given its niche topic and highly technical content. It also feels quite text-heavy.

**Questions:**

Q1. Can we summarize the key points as follows: Under a specific infinite variance prior, the covariance kernel becomes a stochastic process at each layer. Even if the marginal does not exist, sequential sampling is still possible. Furthermore, conditioning on a covariance sample at a given layer yields a Gaussian representation, while marginalizing over the covariance leads the representation to follow a heavy-tailed distribution centered at 0.

Q2. You could perhaps include a 1-dimensional regression example to illustrate the model’s behavior for varying $\alpha$. Additionally, it might be helpful to show the covariance matrix to demonstrate its properties and the correlation among input data.

Q3. Could you clarify the technical differences between the current paper and Loria & Bhadra (2024), which makes the approach feasible?

---

> ### Author Response · Authors · 2024-11-16
> **Response to a3tm**
>
> **Weaknesses:**
>
> The purpose of our paper is to establish the infinite-width limit of deep BNNs under infinite variance priors. This limiting investigation naturally involves appeals to the (generalized) CLT and appropriate results from kernel and Gaussian processes literature. Nevertheless, we have tried to succinctly describe the main innovations in the Introduction in a manner as non-technical as possible, with minimal use of notations.  We have also made every effort to separate out the unavoidable technical parts into theorems, rather than leaving them in the main text. We believe even a non-technical audience might stand to benefit from our main observations and numerical results. We hope you might be able to reconsider your position in the light of our responses to your and the rest of the reviwers' questions and perhaps adjust your score accordingly.
>
> **Questions**
>
> - Q1: Your summary is overall quite accurate, with some caveats. First, we don't need any *specific* infinite variance prior. Indeed, convergence to a stable process would occur under the relatively broad conditions of Theorem 4 in Appendix A, under *any* infinite variance prior distribution (symmetric and zero-centered). Furthermore, it is not that the deep kernel process is our starting point. Rather, a deep infinitely wide BNN with infinite variance prior weights *converges* to this deep kernel process *in distribution*, upon which the results of Cho and Saul (2009) can be leveraged to compute the kernel recursively. This is what we establish in Theorem 1. The conditionally Gaussian representation we derive allows representation learning, an important departure from the GP limit, where the kernel is deterministic. We believe these contributions are succinctly summarized in Section 1.2, but still please let us know if you require further clarifications.
>
> -  Q2:
>
> -- a) We welcome this thoughtful suggestion, and have added a 1-dimensional example with varying $\alpha$s. See  **Figure 6** in Supplementary Section H.3.3, which shows the function fit becoming progressively smoother as $\alpha$ increases closer to 2, the Gaussian limit.
>
> -- b) Given that the *marginal* covariance is not well-defined for stable processes, we need to use the conditional mutual information (see Figure 1) to explore the properties of input dependence. However, following your suggestion, we have added  **Figure 7** in Supplementary Section H.3.4, where we plot the 25th, 50th and 75th percentiles of the posterior of the random covariance kernel in the conditional GP representation for the simulation examples, as another way to visualize representation learning, which is not possible under a deterministic kernel in the GP limit.
>
>
> - Q3: The key technical difference is that Loria and Bhadra (2024) *miss the kernel trick* and attempt to compute the covariance kernel in the feature space using a basis expansion type approach that leads to an  exponential time complexity algorithm of $O(n^{I+2})$, where $I$ is the input dimension and $n$ is the number of data points. Moreover, they never consider a deep network, and restrict themselves to shallow nets. This makes their approach all but infeasible for practical problems, say for $I>2$. In contrast, the key contribution of the current paper is to show both of these challenges (computational complexity and deep networks) can be successfully addressed by working in the *kernel space*. This is a non-trivial, as well as practical advance, over what Loria and Bhadra (2024) managed to accomplish.

---

> > ### Author Response · Authors · 2024-11-26
> >
> > Since we are nearing an end of the time period for making any further changes to the manuscript, we respectfully want to check with Reviewer a3tm if our initial rebuttal was satisfactory and in case there are any pending questions that will help your reassessment.

---

### Author Response · Authors · 2024-11-16
**Overall summary of the changes to the manuscript**

We thank all reviewers for their helpful and constructive feedback. We note that currently a majority of the reviewers are leaning towards acceptance, although several  are on the boundary. We hope the further explanations and changes made to the manuscript are helpful in their re-assessment. Following is a succinct summary of the main changes made in the revised manuscript in response to the comments.

1. Following a request from Reviewers a3tm and xtbp, We have added **Figure 6** that shows the model fit and uncertainty quantification in the 1-d example  as a function of $\alpha$. One can see that the model can capture jumps better at low $\alpha$, and tends to provide a smoother fit as $\alpha\to 2$, the Gaussian limit. This is a nontrivial difference and improvement over the GP regime for the deep $\alpha$ kernel process.

2. We have also added **Figure 7**, where we show the posterior of the *random kernel* learned from the data at 25th, 50th and 75th percentiles. This is a key difference with the GP limit, where the kernel is *deterministic* with a *degenerate posterior*, precluding any possibility of *representation learning*.

3. We are grateful to Reviewer Qsid for an *extremely informative* review, and for anticipating crucial non-trivial future developments that can build on the current work, most notably adapting the current work to the Neural Tangent Kernel (NTK) regime (Jacot et al., 2018, NeurIPS), where the convergence to a GP limit occurs not due to taking a wide limit, but due to the limiting dynamics of the *Gaussian* stochastic gradient noise. Several recent works have found empirical evidence that SGD noise can in fact be non-Gaussian and heavy tailed (e.g., Simsekli et al., 2019, ICML, "A Tail-Index Analysis of Stochastic Gradient Noise in Deep Neural Networks"). In this regime, one should expect a non-Gaussian limit for SGD, similar to ours. We strongly believe these non-trivial future developments should be possible, and will help justify the impact of the current work. **These suggestions for future possibilities have helped us improve the Concluding Remarks Section (Section 5).** We are also grateful to Reviewer Qsid for pointing out some relevant additional references that we have included in the revised paper.

---

### Meta-Review · Area_Chair_Di2D · 2024-12-20

**Metareview:**

This paper explores infinite-width Bayesian neural networks with elliptically-distributed weights with infinite variance, demonstrating convergence to a process with α-stable marginals and conditionally Gaussian representations. This results in a stochastic, data-dependent covariance kernel capable of representation learning, unlike the deterministic kernels arising from finite-variance priors. The authors also provide a computationally efficient MCMC method for inference and demonstrate performance benefits on benchmark datasets, particularly in modeling discontinuous or heavy-tailed data.

The reviewers noted that the paper's presentation is dense and technical, making it challenging for a broader audience, particularly in the abstract and introduction. They also found that the motivation behind the specific model formulation and its advantages over previous approaches could be articulated more clearly, and that the experimental descriptions lack depth and connection to theoretical findings. Nevertheless, the novel perspectives, strong theoretical results, and interesting conclusions suggest that this paper should be published and I recommend acceptance.

**Additional Comments On Reviewer Discussion:**

Some reviewers criticized the paper's dense and technical presentation, particularly the abstract and introduction, noting that it might be difficult to understand the motivation and experimental setup. They also raised specific technical concerns regarding figure clarity, terminology placement, and the need for a more intuitive explanation of the key contributions.

The authors defended the paper's technical nature as inherent to the topic but acknowledged clarity issues, adding visualizations and clarifying technical details. They addressed specific concerns about figures and terminology, emphasized the significance of representation learning in their model, and discussed potential extensions.

After the discussion period, some reviewers increased their scores, and others did not, leaving three reviewers strongly in favor of acceptance, and two weakly opposed. The opposition seems to be low-confidence, and largely based on the dense technical presentation.  Having seen the authors' responses and their revisions, I believe the paper is now sufficiently approachable, would benefit the community, and should be published.

---

### Decision · Program_Chairs · 2025-01-22

Accept (Poster)